# Identification of a third myosin-5a-melanophilin interaction that mediates the association of myosin-5a with melanosomes

Jiabin Pan[1,2], Rui Zhou[1,2], Lin-Lin Yao[1], Jie Zhang[1], Ning Zhang[1], Qing-Juan Cao[1], Shaopeng Sun[1,2], Xiang-dong Li[1,2]*

[1]Group of Cell Motility and Muscle Contraction, State Key Laboratory of Integrated Management of Pest Insects and Rodents, Institute of Zoology, Chinese Academy of Sciences, Beijing, China; [2]University of Chinese Academy of Sciences, Beijing, China

*For correspondence:
lixd@ioz.ac.cn

**Competing interest:** The authors declare that no competing interests exist.

**Abstract** Transport and localization of melanosome at the periphery region of melanocyte are depended on myosin-5a (Myo5a), which associates with melanosome by interacting with its adaptor protein melanophilin (Mlph). Mlph contains four functional regions, including Rab27a-binding domain, Myo5a GTD-binding motif (GTBM), Myo5a exon F-binding domain (EFBD), and actin-binding domain (ABD). The association of Myo5a with Mlph is known to be mediated by two specific interactions: the interaction between the exon-F-encoded region of Myo5a and Mlph-EFBD and that between Myo5a-GTD and Mlph-GTBM. Here, we identify a third interaction between Myo5a and Mlph, that is, the interaction between the exon-G-encoded region of Myo5a and Mlph-ABD. The exon-G/ABD interaction is independent from the exon-F/EFBD interaction and is required for the association of Myo5a with melanosome. Moreover, we demonstrate that Mlph-ABD interacts with either the exon-G or actin filament, but cannot interact with both of them simultaneously. Based on above findings, we propose a new model for the Mlph-mediated Myo5a transportation of melanosomes.

## eLife assessment

This study represents a **useful** description of a third interaction site between melanophilin and myosin-5a which has a role in regulating the distribution of pigment granules in melanocytes. While much of the data forms a **solid** case for this interaction, the inclusion of controls for the cellular studies and measurement of interaction affinities would have been helpful.

## Introduction

Melanosome is a specialized organelle within which melanin pigments are synthesized and stored. The peripheral accumulation of melanosomes in melanocytes is essential for the intercellular transfer of the organelles from the dendrite tips of melanocytes to the adjacent keratinocytes (*Hume and Seabra, 2011*). Microtubule and the associated motor proteins are responsible for the long-range and bidirectional transport of melanosomes between the nucleus to cell periphery, and actin filament and the associated motor protein myosin-5a (Myo5a) are essential for the short-distance transport and the capture of melanosomes at cell periphery (*Wu et al., 1998*; *Wu and Hammer, 2014*). The overall effect of the two transport systems is the peripheral accumulation of melanosomes in melanocytes.

Lack of the latter causes the perinuclear accumulation of melanosomes in melanocytes, a phenotype called 'dilute' (*Provance et al., 1996*; *Wei et al., 1997*).

Myo5a is a processive motor that has been implicated in the transportation and the tethering of organelles along actin filaments (*Lindsay and McCaffrey, 2014*; *Mehta et al., 1999*; *Rudolf et al., 2011*; *Yoshimura et al., 2006*; *Zhang et al., 2018*). Myo5a has two heavy chains, each containing an N-terminal motor domain, six IQ motifs that act as the binding sites for calmodulin (CaM) or CaM-like light chains, and a tail (*Ikebe, 2008*; *Zhang et al., 2018*). The tail of Myo5a can be further divided into three segments. The proximal tail is a long coiled-coil with length about 200 residues. The distal tail is the C-terminal globular tail domain (GTD). Between the proximal tail and the GTD is the middle tail, which is comprised of several short coiled-coils and the connecting loops. At least three essential functions are served by Myo5a tail. First, the coiled-coil regions enable Myo5a to form a dimer. Second, the GTD inhibits the motor activity of myosin head. Third, the tail is the binding site for the cargo proteins. The motor activity of Myo5a is tightly regulated (*Krementsov et al., 2004*; *Li et al., 2006*; *Li et al., 2008*; *Li et al., 2004*; *Liu et al., 2006*; *Thirumurugan et al., 2006*; *Wang et al., 2004*). In the inhibited state, Myo5a forms a folded conformation, in which the GTDs fold back to interact and inhibit the motor function. Upon being activated by high $Ca^{2+}$, cargo or adaptor proteins, Myo5a transforms from the folded conformation to the extended conformation, in which the GTD dissociates from the motor domain and the inhibition is relieved.

Melanophilin (Mlph) is one of the best-studied cargo proteins of Myo5a. Together with Rab27a, Mlph mediates the attachment of Myo5a to melanosomes. Rab27a is a small GTPase that, in the GTP-bound form, is anchored into the melanosome membrane via its lipid moiety (*Ishida et al., 2014*; *Wandinger-Ness and Zerial, 2014*) and Mlph bridges the interaction between Rab27a and Myo5a (*Fukuda et al., 2002b*; *Nagashima et al., 2002*; *Wu et al., 2002a*). The tripartite complex of Rab27a, Mlph, and Myo5a connects the melanosome to the actin filament. Lacking any one of these three components in melanocytes causes dilute phenotype perinuclear accumulation of melanosomes (*Fukuda et al., 2002b*; *Hume et al., 2001*; *Strom et al., 2002*; *Wei et al., 1997*; *Wu et al., 2001*; *Wu et al., 2002a*).

Mlph contains four functional regions, including RBD (Rab27a-binding domain), GTBM (Myo5a GTD-binding motif), EFBD (Myo5a exon F-binding domain), and ABD actin-binding domain (*Fukuda et al., 2002b*; *Geething and Spudich, 2007*; *Li et al., 2005*; *Yao et al., 2015*). The association of Myo5a with Mlph involves two molecular interactions. One is the interaction between the exon-F-encoded region (exon-F for simplification) of Myo5a and the EFBD of Mlph (*Fukuda and Itoh, 2004*; *Wu et al., 2002b*; *Wu et al., 2002a*). Exon-F is an alternatively spliced exon presents in the middle tail of the melanocyte-spliced isoform of Myo5a (*Huang et al., 1998*). The exon-F/EFBD interaction is absolutely essential for the localization of Myo5a to the melanosome (*Wu et al., 2002b*). The other interaction is between the GTD of Myo5a and the GTBM of Mlph (*Geething and Spudich, 2007*; *Pylypenko et al., 2013*; *Wei et al., 2013*). We previously found that GTBM activates Myo5a motor function by relieving the GTD inhibition on the motor domain and inducing a folded-to-extended conformational transition of Myo5a (*Li et al., 2005*; *Yao et al., 2015*). Recently, we demonstrated that the activation of Myo5a by Mlph-GTBM is regulated by another Myo5a-binding protein RILPL2 (Rab-interacting lysosomal protein-like 2) and the small GTPase Rab36, a binding partner of RILPL2 (*Cao et al., 2019*).

Mlph-ABD is essential to proper melanosome transport (*Fukuda and Kuroda, 2002a*; *Kuroda et al., 2003*). Besides being able to bind to actin filament, Mlph-ABD could interact with end-binding protein 1, which might facilitate the transfer of melanosomes from microtubules to actin (*Wu et al., 2005*). In vitro motility assays at single-molecular level show that Mlph prolongs and slows the processive movements of Myo5a, presumably via the interaction between ABD and actin filament (*Sckolnick et al., 2013*). A cluster of conserved positively charged residues in ABD is essential for actin binding and Melanophilin localization in melanocytes (*Kuroda et al., 2003*). An in vitro study suggests that Mlph-ABD is a target of protein kinase A and that the phosphorylated Mlph preferred binding to actin even in the presence of microtubules, whereas dephosphorylated Mlph bind to microtubules (*Oberhofer et al., 2017*).

Biochemical analysis of Mlph show that the ABD is not essential for the interaction with Myo5a nor the activation of Myo5a motor function, suggesting that a truncated Mlph protein containing the RBD and the two Myo5a-binding domains (the GTBM and the EFBD) but lacking the ABD is able to form a

tripartite complex with Rab27a and Myo5a and thus sufficient for the recruitment of Myo5a to mela-nosome membrane (*Kuroda et al., 2003*; *Wu et al., 2002b*). However, using melan-ln cells (Mlph-null melanocytes), Hume et al. have found that the ABD-deleted Mlph cannot to promote the recruitment of Myo5a onto melanosomes, indicating that the ABD is necessary in situ for the association of Myo5a with Mlph (*Hume et al., 2006*). This discrepancy between in vitro and cell culture results promoted us to investigate the interaction between Mlph-ABD and Myo5a.

In this study, we have identified a third interaction between Myo5a and Mlph, that is, the interac-tion between the exon-G-encoded region (exon-G for simplification) of Myo5a and Mlph-ABD. We find that the exon-G/ABD interaction is independent from the exon-F/EFBD interaction, and that, similar to exon-F/EFBD, exon-G/ABD is essential for the dilute-like phenotype induced by Myo5a-tail overexpression. Moreover, we demonstrate that Mlph-ABD interacts with either the exon-G of Myo5a or actin filament, but cannot interact with both of them simultaneously. Based on above findings, we propose a new model for the Mlph-mediated Myo5a transportation of melanosomes.

## Results

### The actin-binding domain of melanophilin (Mlph-ABD) contains a novel Myo5a-binding site

Myo5a binds to its cargo proteins, including Mlph, majorly via the middle tail domain (MTD, residues 1106–1467) and the GTD (residues 1468–1877; *Figure 1A*, top). The C-terminal portion of the MTD is encoded by a serial exons, called exon-A to -G, among which exon-B, -D, and -F are alternatively spliced exons (*Seperack et al., 1995*). The melanocyte isoform of Myo5a contains exon-D and -F (*Figure 1A*, middle), whereas the brain isoform contains exon-B. The MTD is predicted to form four short coiled-coils, among which the last one is formed by the C-terminus of exon-F and the entire exon-G (*Figure 1A*, bottom).

To detect interaction between the ABD of Mlph with Myo5a, we performed GST pulldown assay using GST-Mlph-ABD (the GST-tagged ABD of Mlph, residues 401–590) and Flag-Myo5a-MTD (the Flag-tagged middle tail domain of Myo5a, residues 1106–1467; *Figure 1A and B*). GST-Mlph-ABD was expressed in *E. coli* and purified by GSH-Sepharose chromatography, and Flag-Myo5a-MTD was expressed in Sf9 cells and purified by anti-Flag affinity chromatography. GST pulldown assay shows that GST-Mlph-ABD specifically interacted with Flag-Myo5a-MTD (*Figure 1C*, lane 2). Deleting the residues 401–437 of Mlph-ABD had little effect on the interaction with Myo5a-MTD (*Figure 1C*, lane 3 and 4), indicating the N-terminal 37 residues of Mlph-ABD are not essential for the interaction. Further deletion of 437–446 slightly decreased the interaction and deletion of 446–455 eliminated the inter-action (*Figure 1C*, lane 5 and 6), indicating that residues 446–455 are essential for the interaction with Myo5a-MTD. Similarly, we performed C-terminal truncation analysis of Mlph-ABD on the interaction with Myo5a-MTD and identified residues 560–571 essential for the interaction (*Figure 1D*). Taking together, we identified the middle portion of Mlph-ABD (residues 446–571) as the core of the novel binding site for Myo5a-MTD.

We then produced a number of truncated Myo5a-MTD constructs and performed GST pulldown assays to narrow down the region in Myo5a-MTD binding to Mlph-ABD (*Figure 2A*). To simplify the experiments, we first compared the bacterial-expressed Myo5a-MTD with the Sf9 cell-expressed Myo5a-MTD in the interaction with Mlph-ABD. GST pulldown assay shows that both versions of Myo5a-MTD specifically interacted with Mlph-ABD (*Figure 2—figure supplement 1A*). All the following pulldown experiments were performed using the bacterial-expressed Myo5a-MTD, unless otherwise indicated. C-terminal truncation of Flag-Myo5a-MTD shows that the C-terminal half of exon-G (residues 1453–1467) is essential for binding to Mlph-ABD (*Figure 2—figure supplement 1B*). Mlph-ABD interacted strongly with Flag-Trx-Myo5a (1411–1467) (a short peptide containing exon-F and exon-G), but weakly with Flag-Trx-Myo5a (1436–1467) (a short peptide containing exon-G) (*Figure 2—figure supplement 1C*, lane 3 and 4), suggesting that exon-F might be required for strong interaction of exon-G with Mlph-ABD. However, deletion of exon-F from Myo5a-MTD did not affect the interaction with Mlph-ABD (*Figure 2B*, lane 2), indicating that exon-F is not essential for the interaction. Deletion of the C-terminal portion (residues 1454–1467) of exon-G abolished the binding of Myo5a-MTD with Mlph-ABD (*Figure 2B*, lane 3), but did not affect the interaction of Myo5a-MTD with Mlph-EFBD (*Figure 2C*). *Figure 2A* summaries the truncation analysis of the interaction between

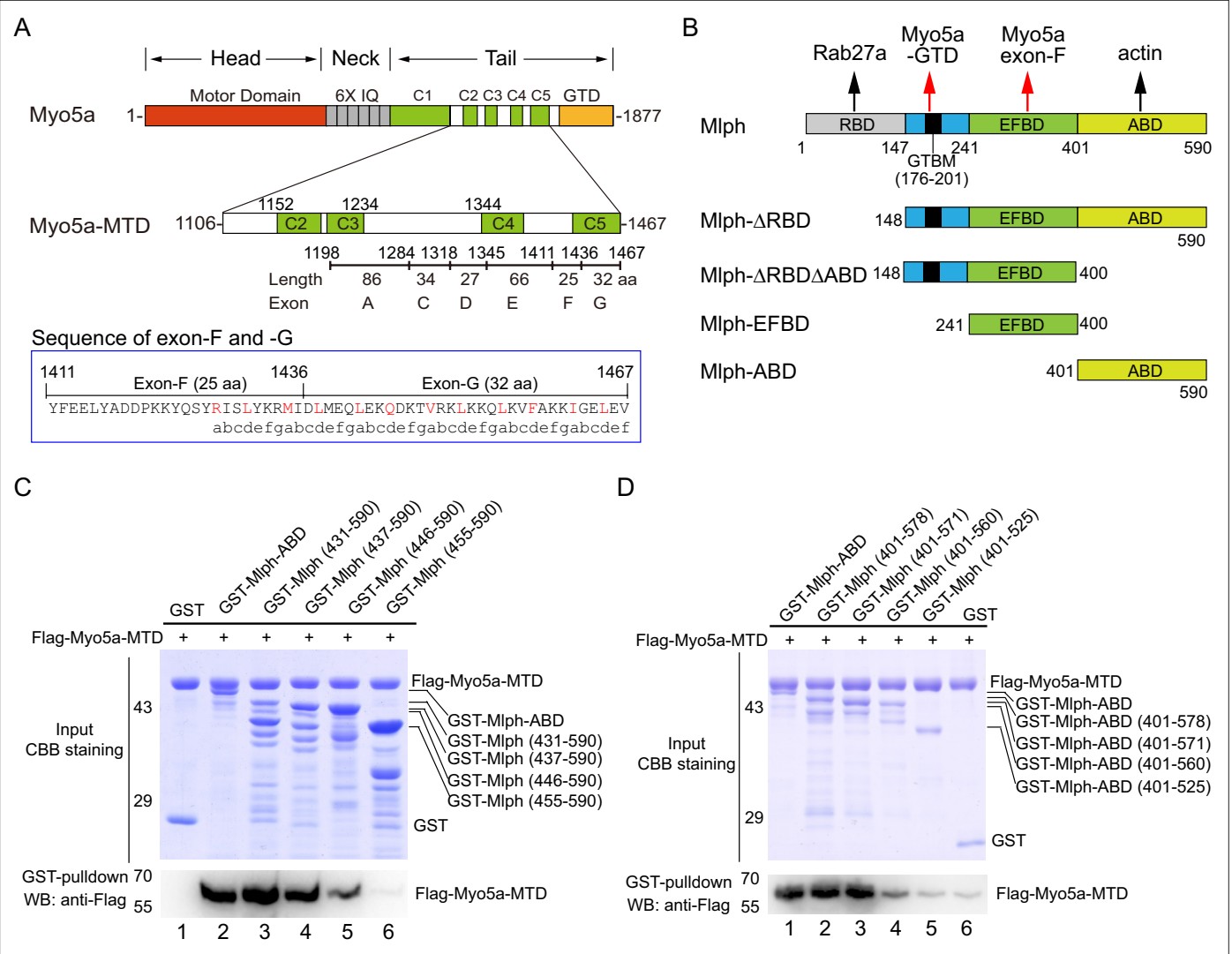

**Figure 1.** The actin-binding domain of melanophilin (Mlph-ABD) interacts with the middle tail domain (MTD) of Myo5a. (**A**) Diagram of the melanocyte-spliced isoform of Myo5a. Myo5a-MTD, Myo5a coiled-coils (residues 1106–1467). IQ, the CaM binding site; GTD, the C-terminal globular tail domain. In blue box is the sequence of exon-F and -G, annotated with the predicted coiled-coil heptad repeats. (**B**) Diagram of melanophilin (Mlph). RBD, Rab27a-binding domain; GTBM, globular tail domain-binding motif; EFBD, exon-F binding domain; ABD, actin-binding domain. (**C, D**) GST pulldown of GST-Mlph-ABD with the N-terminus truncated (**C**) or the C-terminus truncated (**D**) with Flag-Myo5a-MTD. GST-Mlph-ABD constructs were bound to GSH-Sepharose and then incubated with Flag-Myo5a-MTD. The GSH-Sepharose-bound proteins were eluted by GSH and analyzed by Western blot using anti-Flag antibody. The inputs were analyzed with SDS-PAGE and visualized by Coomassie brilliant blue (CBB) staining. GST was used as negative control. Note: The Sf9 cell-expressed Myo5a-MTD was used in the GST pulldown assays shown in this figure.

The online version of this article includes the following source data for figure 1:

**Source data 1.** Original and uncropped gels and blots for *Figure 1C*.

**Source data 2.** Original and uncropped gels and blots for *Figure 1D*.

Myo5a-MTD and Mlph-ABD. Based on above results, we conclude that the exon-G of Myo5a binds to Mlph-ABD.

## Characterization of the exon-G/ABD interaction

To characterize the exon-G/ABD interaction, we monitored the interaction between Myo5a-MTD and Mlph-ABD using microscale thermophoresis (MST), and obtained the dissociation constant ($K_d$) of 562±169 nM of Myo5a-MTD for binding to Mlph-ABD (*Figure 3A*). Consistent with the pulldown assay (*Figure 2B*), deletion of the C-terminal half of exon-G (Myo5a-MTDΔG) greatly decreased the

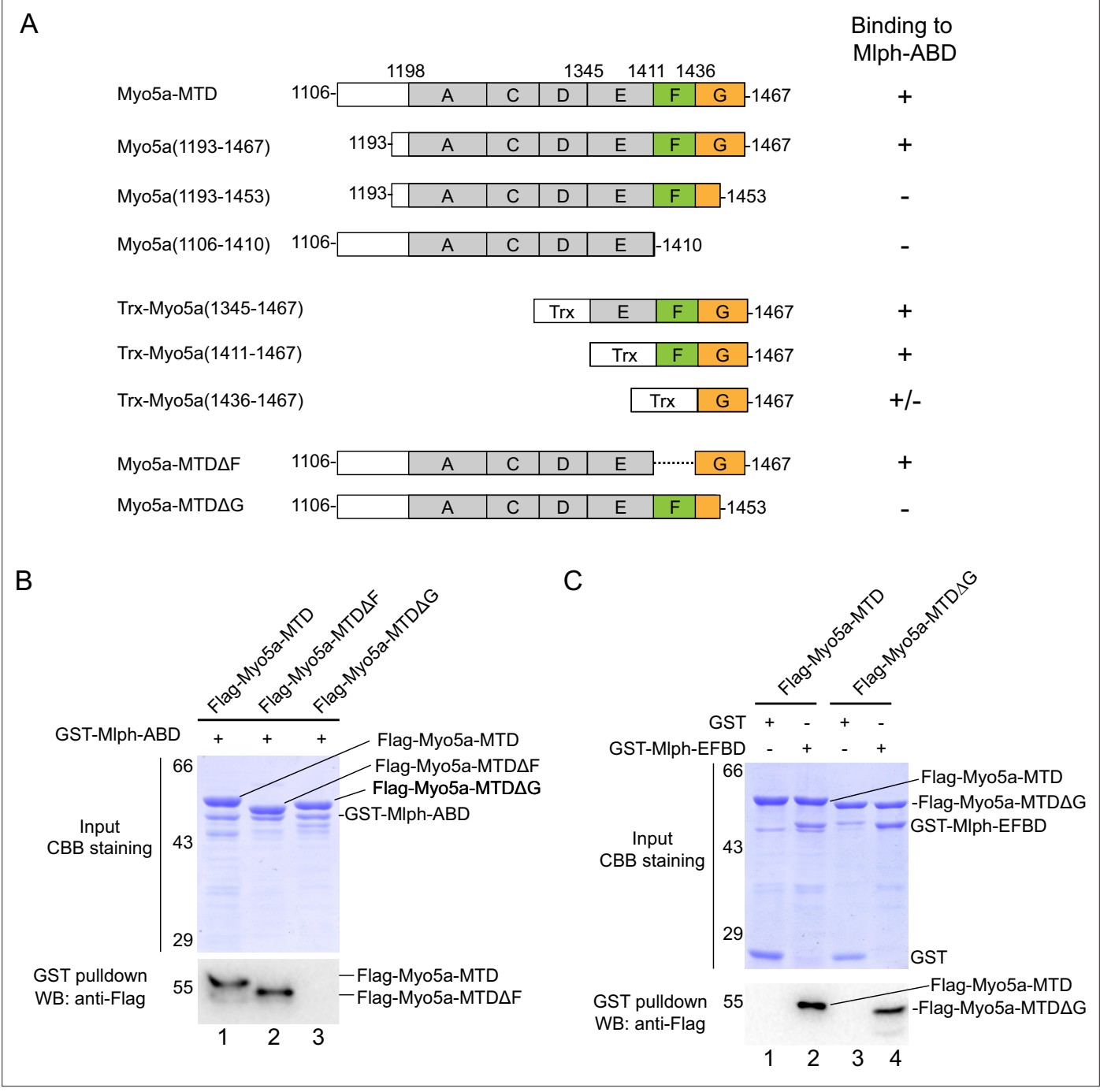

**Figure 2.** The exon-G of Myo5a interacts with Mlph-ABD. (**A**) Summary of the interactions between the truncated Myo5a constructs and Mlph-ABD based on the GST pulldown assay shown in *Figure 2—figure supplement 1A*. +, strong interaction; +/-, weak interaction; -, no interaction. (**B**) Deletion of the C-terminal portion of exon-G abolishes the interaction between Myo5a-MTD and Mlph-ABD. (**C**) Deletion of the C-terminal portion of exon-G does not affect the interaction between Myo5a-MTD and Mlph-EFBD. GST pulldown assays were performed using GST-Mlph-ABD and Flag-Myo5a-MTD variants. The GSH-Sepharose-bound proteins were eluted by GSH and analyzed by Western blot using anti-FLAG antibody; the inputs were analyzed with SDS-PAGE and visualized by Coomassie brilliant blue (CBB) staining.

The online version of this article includes the following source data and figure supplement(s) for figure 2:

**Source data 1.** Original and uncropped gels and blots for *Figure 2B*.

**Source data 2.** Original and uncropped gels and blots for *Figure 2C*.

*Figure 2 continued on next page*

*Figure 2 continued*

**Figure supplement 1.** Identification of the Mlph-ABD-binding site in the middle tail of Myo5a.GST pulldown assays were performed using GST-Mlph-ABD and Flag-Myo5a-MTD variants.

**Figure supplement 1—source data 1.** Original and uncropped gels and blots for *Figure 2—figure supplement 1A*.

**Figure supplement 1—source data 2.** Original and uncropped gels and blots for *Figure 2—figure supplement 1B*.

**Figure supplement 1—source data 3.** Original and uncropped gels and blots for *Figure 2—figure supplement 1C*.

MST signaling (*Figure 3A*). We then investigated the effect of ionic strength on the exon-G/ABD interaction. GST pulldown assays of GST-Mlph-ABD and Flag-Myo5a-MTD were conducted in the presence different concentrations of NaCl. Compared with 100 mM NaCl, 300 mM and 500 mM NaCl greatly decreased the amount of Flag-Myo5a-MTD pulled down with GST-Mlph-ABD (*Figure 3B*), suggesting that ionic interactions are critical for the exon-G/ABD interaction.

Sequence alignment among the Mlph of mammals and birds reveals several highly conserved charged residues in the two regions of Mlph-ABD essential for the interaction with Myo5a-MTD, including four acidic residues (D447, E449, E452, and E454) in the residues 446–455 and two basic residues (R562 and R568) in the residues 560–571 (*Figure 3C*). To determine the roles of those conserved charged residues of Mlph on the exon-G/ABD interaction, we mutated them individually into alanine. GST pulldown assays show that both E449A and E452A mutations abolished the interaction with Myo5a-MTD, whereas other mutations (D447A, E454A, R562A, R568A) had little effect on the interaction (*Figure 3E*), indicating that both E449 and E452 of Mlph are essential for the exon-G/ABD interaction.

We expected the conserved acidic residues E449 and E452 of Mlph to interact with the conserved basic residues in the exon-G of Myo5a. Sequence alignment of the three Myo5 isoforms of mammals and birds reveals two highly conserved basic residues (K1456 and K1460) in the C-terminal portion of exon-G of Myo5a (*Figure 3D*). GST pulldown assay shows that both K1456A and K1460A mutations substantially decreased the amount of Flag-Myo5a-MTD pulled down with GST-Mlph-ABD (*Figure 3F*), suggesting that both K1456 and K1460 of Myo5a play a key role in the interaction with Mlph-ABD.

## The exon-F/EFBD interaction and the exon-G/ABD interaction are independent from each other

Because exon-F and exon-G are in immediate vicinity, it is possible that the exon-F/EFBD interaction and the exon-G/ABD interaction interfere with each other. To test this possibility, we performed GST pulldown assays of GST-Mlph-ABD and Flag-Mlph-EFBD in the presence or absence of Flag-Myo5a-MTD. If Myo5a-MTD is able to interact with both Mlph-ABD and Mlph-EFBD simultaneously, a tripartite complex will be formed by Myo5a-MTD, Mlph-ABD, and Mlph-EFBD (*Figure 4A*, upper panel), and Flag-Mlph-EFBD will be pulled down with GST-Mlph-ABD in the presence of Flag-Myo5a-MTD. As shown in *Figure 4A*, Flag-Mlph-EFBD was pulled down with GST-Mlph-ABD in the presence of Flag-Myo5a-MTD, but not in the absence of Flag-Myo5a-MTD. This result is consistent with a scenario that Myo5a-MTD binds to both Mlph-EFBD and Mlph-ABD simultaneously. We therefore conclude that the exon-F/EFBD interaction and the exon-G/ABD interaction do not interfere with each other.

As the exon-F/EFBD interaction and the exon-G/ABD interaction are independent to each other, we expected that those two interactions act synergically for the binding of Myo5a-MTD to Mlph. As expected, GST pulldown assays show that GST-Mlph-ΔRBD, which contains both EFBD and ABD, strongly bound to Flag-Myo5a-MTD, and deletion of either exon-F or exon-G from Myo5a-MTD substantially weakened the interaction with GST-Mlph-ΔRBD (*Figure 4B*). Conversely, Flag pulldown shows that substantial amount of GST-Mlph-ΔRBD could be pulled down with Flag-Myo5a-MTD, and deleting ABD from GST-Mlph-ΔRBD strongly decreased the amount of protein pulled down with Flag-Myo5a-MTD (*Figure 4C*). Above results indicate that the exon-F/EFBD interaction and the exon-G/ABD interaction act synergically, resulting in a strong interaction between Myo5a-MTD and Mlph.

## Mlph-ABD cannot interact with F-actin and exon-G simultaneously

In addition to interacting with the exon-G of Myo5a, Mlph-ABD is able to bind to actin. To investigate whether Mlph-ABD interacts with actin and exon-G simultaneously, we performed actin

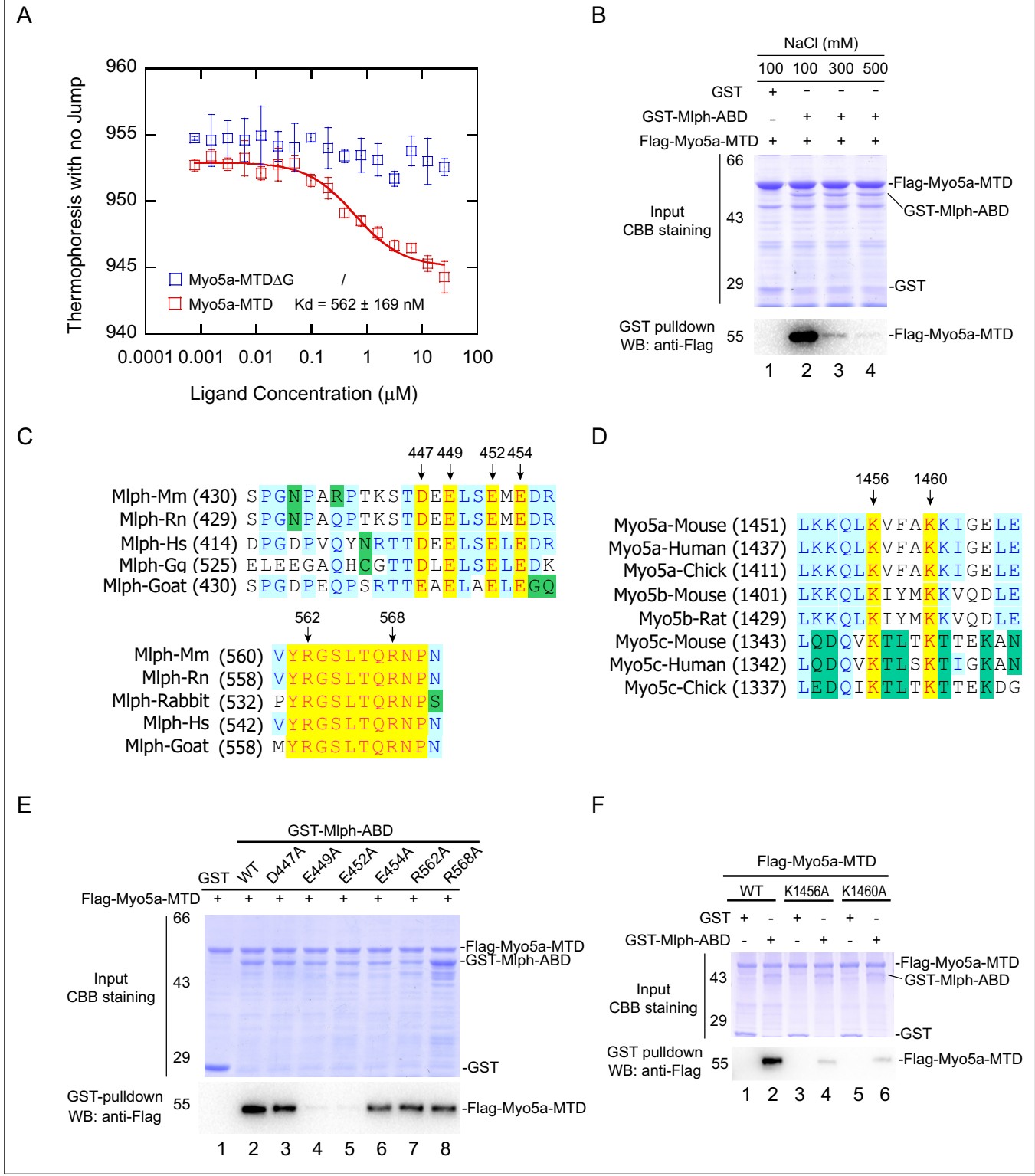

**Figure 3.** Identification of the key residues for the exon-G/ABD interaction. (**A**) Dissociation constant ($K_d$) of Myo5a-MTD or Myo5a-MTDΔG binding to Mlph-ABD measured by MST. The solid curve was fit to the standard $K_d$-fit function. Results are presented as the mean ± SEM of three independent experiments. (**B**) Ionic strength dependence of the interaction between Myo5a-MTD and Mlph-ABD. GST pulldown was performed using GST-Mlph-ABD and Flag-Myo5a-MTD in the presence of different concentrations of NaCl. (**C**) Sequence alignments of the regions in Mlph-ABD essential for binding to Myo5a-MTD. Conserved charged residues are indicated. (**D**) Sequence alignments of the C-terminal portion of exon-G, an essential region for binding to Mlph-ABD. Two conserved basic residues (K1456 and K1460) are indicated. (**E**) Effects of alanine mutation of the conserved charged

*Figure 3 continued on next page*

*Figure 3 continued*

residues in Mlph-ABD on the interaction with Myo5a-MTD. E449A and E452A mutations in Mlph abolished the interaction between Mlph-ABD and Myo5a-MTD. (**F**) Effects of K1456A and K1460A mutations of Myo5a-MTD on the interaction with Mlph-ABD.

The online version of this article includes the following source data for figure 3:

**Source data 1.** Original MST data for *Figure 3A*.

**Source data 2.** Original and uncropped gels and blots for *Figure 3B*.

**Source data 3.** Original and uncropped gels and blots for *Figure 3E*.

**Source data 4.** Original and uncropped gels and blots for *Figure 3F*.

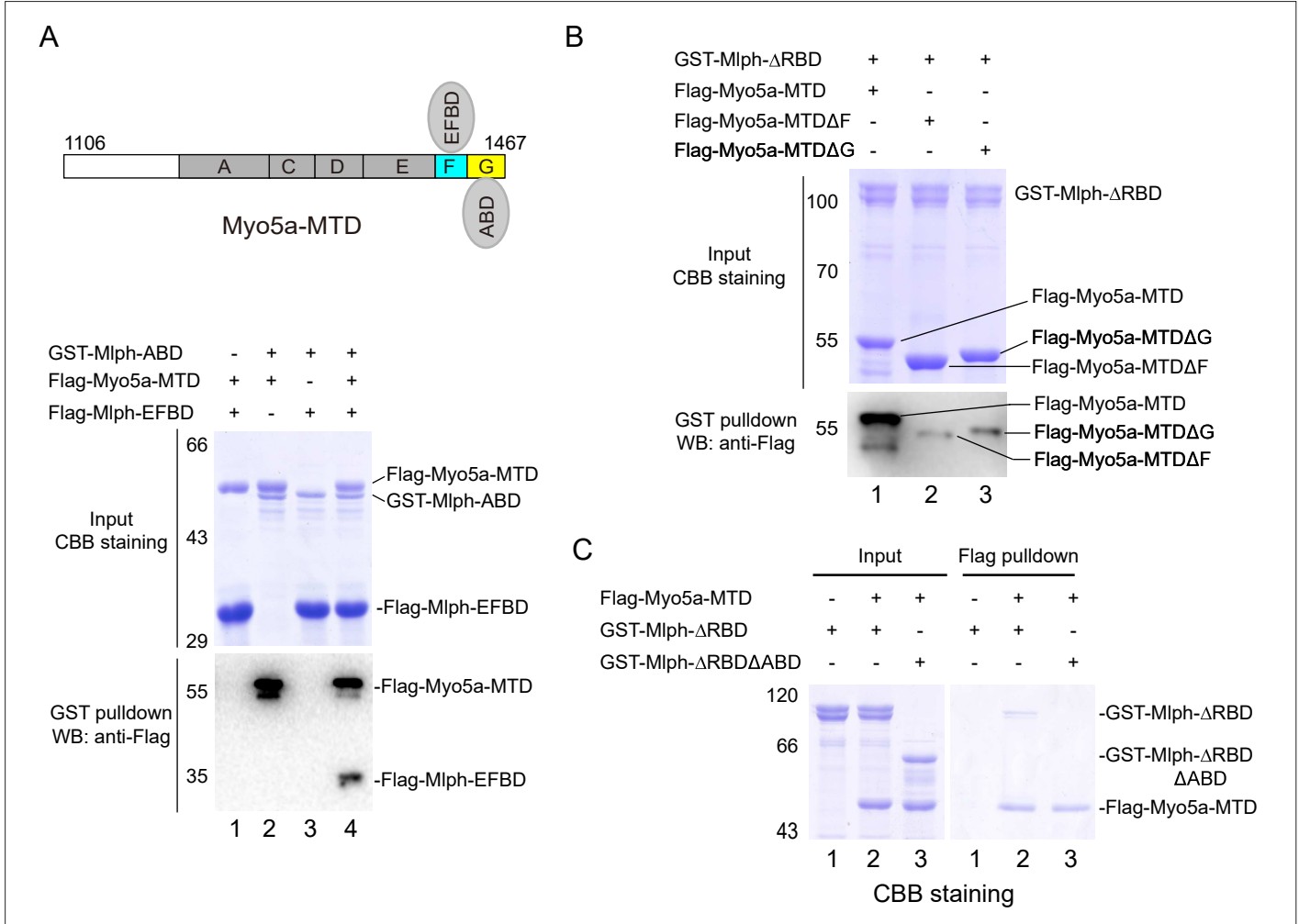

**Figure 4.** The exon-F/EFBD and the exon-G/ABD interactions act synergistically. (**A**) Myo5a-MTD bridges between Mlph-ABD and Mlph-EFBD. Upper, diagram shows Myo5a-MTD binds to both the EFBD and the ABD of Mlph. Lower, GST pulldown assays of GST-Mlph-ABD and Flag-Mlph-EFBD with or without Flag-Myo5a-MTD. (**B**) Both exon-F and exon-G of Myo5a-MTD are required for the strong interaction with Mlph-ΔRBD. GST pulldown of GST-Mlph-ΔRBD with Flag-Myo5a-MTD variants. (**C**) ABD is essential for the strong interaction between Mlph-ΔRBD and Myo5a-MTD. The input samples were analyzed by SDS-PAGE and visualized by CBB staining. The pulled down samples were analyzed by western blot using anti-Flag antibody (**A** and **B**) or by SDS-PAGE with CBB staining (**C**).

The online version of this article includes the following source data for figure 4:

**Source data 1.** Original and uncropped gels and blots for *Figure 4A*.

**Source data 2.** Original and uncropped gels and blots for *Figure 4B*.

**Source data 3.** Original and uncropped gels for *Figure 4C*.

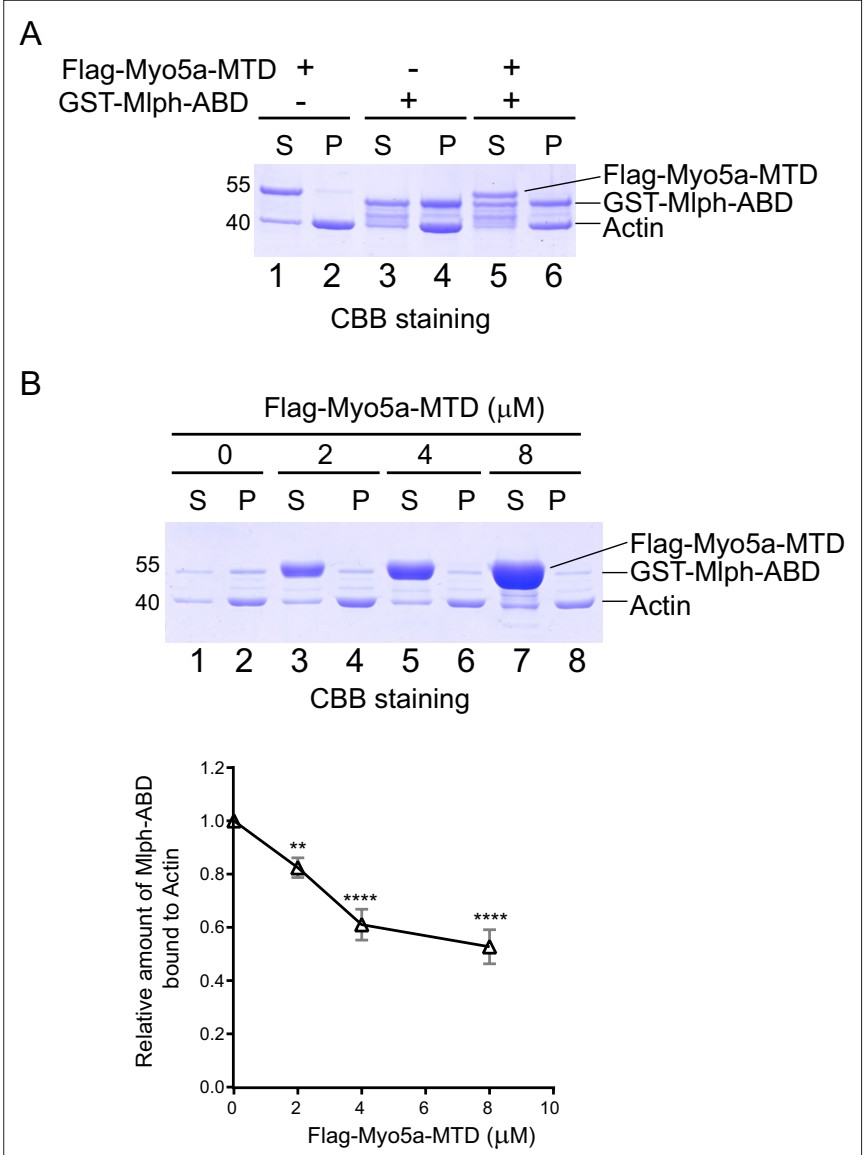

**Figure 5.** Myo5a-MTD antagonizes the interaction between Mlph-ABD and actin. (**A**) GST-Mlph-ABD and/or Flag-Myo5a-MTD were incubated with actin and then subjected to ultracentrifugation. The supernatants (S) and the pellets (P) were analyzed by SDS-PAGE (10%) with CBB staining. (**B**) GST-Mlph-ABD was incubated with actin in the presence of different concentrations of Flag-Myo5a-MTD and then subjected to to ultracentrifugation. Upper panel, the supernatants (S) and the pellets (P) were analyzed by SDS-PAGE (10%) with CBB staining. Lower panel, the amounts of GST-Mlph-ABD co-sedimentated with actin in the presence of different concentration of Flag-Myo5a-MTD were quantified based on the density in the SDS-PAGE. Data are the mean ± SD of three independent experiments with one-way ANOVA with post-hoc Bonferroni test. **$p<0.01$, ****$p<0.0001$. Note: a Sf9 cell-expressed Myo5a-MTD was used in the actin cosedimentation assays shown in this figure.

The online version of this article includes the following source data for figure 5:

**Source data 1.** Original and uncropped gels for *Figure 5A*.

**Source data 2.** Original and uncropped gels and statistical data for *Figure 5B*.

co-sedimentation of Mlph-ABD and Myo5a-MTD. As expected, substantial amount of Mlph-ABD, but essentially no Myo5a-MTD, was co-sedimented with F-actin (*Figure 5A*). Interestingly, Mlph-ABD did not increase the amount of Myo5a-MTD co-sedimented with F-actin (*Figure 5A*), suggesting that Mlph-ABD cannot interact with F-actin and exon-G simultaneously.

The inability of Mlph-ABD to interact with F-actin and exon-G simultaneously suggests that F-actin and exon-G compete in binding to Mlph-ABD. It is possible that Myo5a-MTD might antagonize the interaction between Mlph-ABD and F-actin. To test this possibility, we performed F-actin co-sedimentations of Mlph-ABD in the presence of different concentrations of Myo5a-MTD. As expected, Myo5a-MTD substantially decreased the amount of Mlph-ABD co-sedimented with F-actin (*Figure 5B*). We therefore concluded that Mlph-ABD cannot bind to F-actin and exon-G simultaneously and those two interactions are mutually exclusive.

## The exon-G/ABD interaction is essential for Myo5a-tail to induce dilute-like phenotype in melanocytes

Myo5a associates with melanosome via its interaction with Mlph, which attaches to melanosome via the interaction between its RBD and the melanosome-bound Rab27a. Overexpression of the EGFP fusion protein of Mlph-ΔRBD in the melanocyte cell line, melan-a, caused dilute-like phenotype, that is the perinuclear distribution of melanosomes in melanocyte (*Figure 6A*; *Figure 6—figure supplement 1*). Generation of dilute-like phenotype by Mlph-ΔRBD was attributed to the ability of Mlph-ΔRBD to compete with the melanosome-bound Mlph in interacting with Myo5a molecules, thereby dissociating Myo5a from melanosome. To investigate whether the exon-G/ABD interaction is essential for the dominant negative effect of Mlph-ΔRBD in melanocyte, we introduced two single pointed mutations, E449A and E452A, in Mlph-ΔRBD, both of which are essential for Mlph-ABD to interact with the exon-G (*Figure 3E*). We found that both two mutations substantially decreased the number of the transfected cells with dilute-like phenotype. While dilute-like phenotype was presented in 48.78% of cells expressing the wild-type Mlph-ΔRBD, only in 20.97% and 19.07% of cells expressing E449A and E542A, respectively (*Figure 6B*). Western blots of the lysate of transfected cells showed no degradation of the expressed proteins of EGFP-Mlph-ΔRBD WT and two mutants (*Figure 6C*). Based on the band densities in the Western blot and the transfection efficiencies of those three constructs, we estimated the relative protein expression levels of each transfected cells for EGFP-Mlph-ΔRBD WT, E449A, and E452A were roughly equal (*Figure 6C*). Therefore, we conclude that the inability of E449A and E452A mutants to generate dilute-like phenotypes was not due to the intracellular degradation or the low expression level of the EGFP fusion proteins.

To further determine the role of the exon-G/ABD interaction on the tethering of Myo5a with Mlph in melanocyte, we expressed the EGFP fusion protein of Myo5a-tail (residues 1106–1877), which contains all three Mlph-binding regions, i.e., exon-F, exon-G, and GTD (*Figure 7A*). Overexpressing the tail region of Myo5a in melanocyte was expected to compete with endogenous Myo5a in interacting with Mlph, thus causing dilute-like phenotype (*Wu et al., 2002a*). Deletion analysis of the tail region of Myo5a have shown that both exon-F and the GTD are required and that neither of them alone is sufficient for generating the dominant negative phenotype in melanocytes (*Wu et al., 2002a*). Similar to the previous report, Myo5a-tail is well co-localized with both melanosome and Mlph, and generated dilute-like phenotype in ~50% of transfected melan-a melanocytes (*Figure 7B and C*). In contrast, Myo5a-TailΔG generated dilute-like phenotype in only ~10% of transfected cells (*Figure 7B and C*; *Figure 7—figure supplement 1*). Western blot of the lysate of transfected cells showed no degradation of the expressed proteins of EGFP-Myo5a-Tail WT and ΔG (*Figure 7D*). Based on the band densities in the western blot and the transfection efficiencies, we estimated the relative protein expression levels of each transfected cells for EGFP-Myo5a-Tail WT and ΔG were roughly equal. Therefore, we conclude that the inability of EGFP-Myo5a-Tail ΔG to generate dilute-like phenotypes was not due to the intracellular degradation or the low expression level of the EGFP fusion protein.

Based on above results, we concluded that, similar to exon-F, exon-G is also required for Myo5a-Tail to disrupt endogenous Myo5a function in transportation of melanosomes.

## Myo5a-MTD colocalized with Mlph but was unable to induce dilute-like phenotype

Unlike Myo5a-Tail, overexpression of Myo5a-MTD, which contains both exon-F and exon-G but lacks the GTD, did not generate dilute-like phenotype in melan-a cells. However, Myo5a-MTD is well colocalized with endogenous Mlph and partially colocalized with melanosome (*Figure 8A*). We reasoned that the presence of both exon-F and exon-G is insufficient for binding to the Mlph occupied by Myo5a, but sufficient for binding to the unoccupied Mlph.

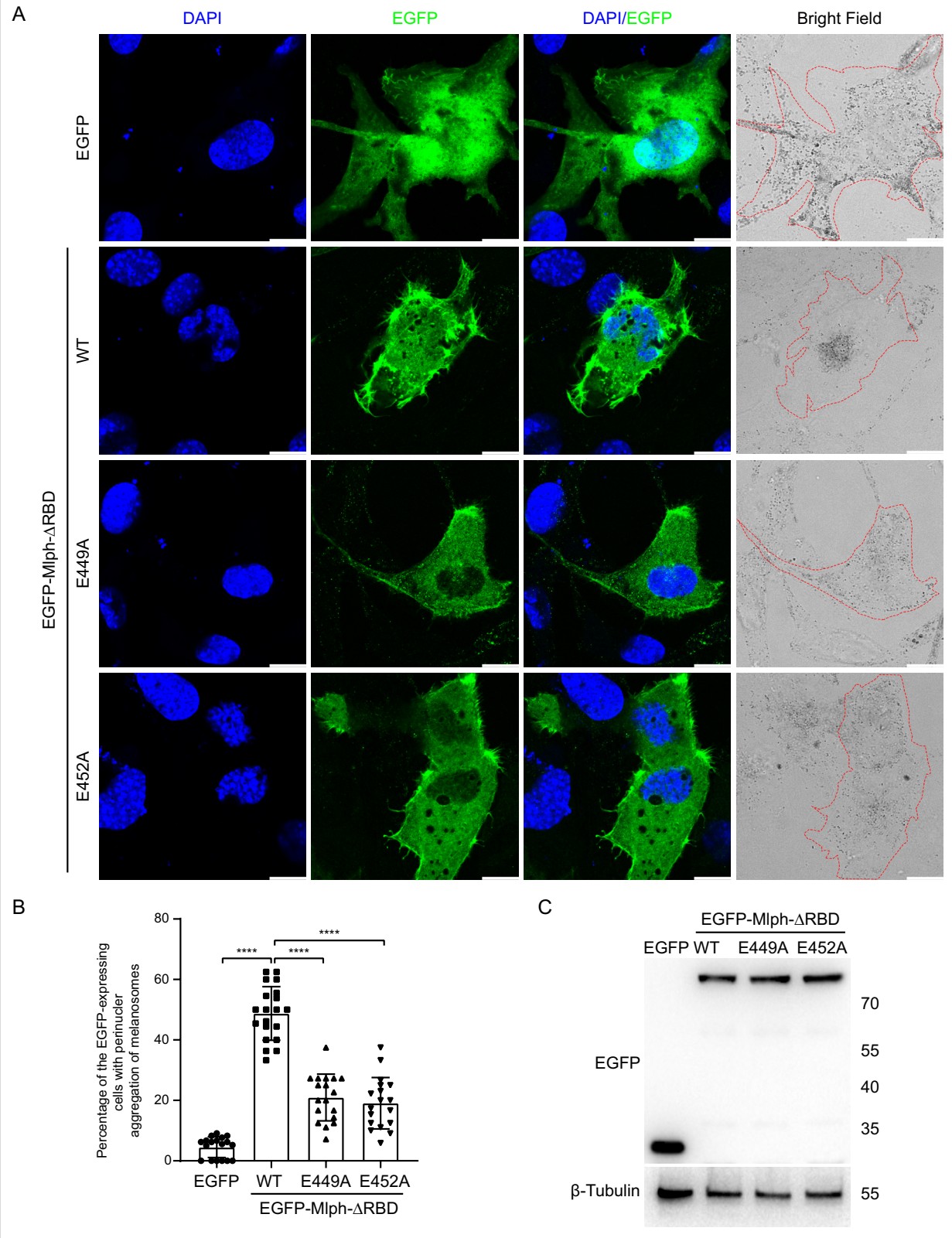

**Figure 6.** Both E449 and E452 of Mlph are essential for the perinuclear distribution of melanosomes induced by Mlph-ΔRBD overexpression. Melan-a melanocytes were transfected to express EGFP-Mlph-ΔRBD (WT, E449A, or E452A) or EGFP. The distribution of melanosomes in the transfected melanocytes was imaged and the number of melanocytes with perinuclear distribution of melanosomes was counted. (**A**) Typical images of melanocytes expressing EGFP alone, EGFP-Mlph-ΔRBD WT, E449A, or E452A. Zooms are x 3. Cells are outlined with red dashed lines. Scale bars = 10 μm. The

*Figure 6 continued on next page*

*Figure 6 continued*

transfection efficiencies were 11% for EGFP (n=275), 9.2% for EGFP-Mlph-ΔRBD WT (n=194), 11.2% for E449A (n=208), and 11.8% for E452A (n=267). n represents cell number. (**B**) The percentage of melanocytes exhibiting perinuclear melanosome aggregation among the transfected melanocytes. Results are presented as the mean ± SD of three independent experiments with one-way ANOVA with post-hoc Bonferroni test. Each point represents the percentage of perinuclear distribution of EGFP fusion proteins expressing cells in each fields. ****p<0.0001. (**C**) Western blot of the lysates of melan-a transfected with EGFP-Mlph-ΔRBD WT, E449A, or E452A. The expressed EGFP fusion proteins were probed with the antibody against EGFP. The relative band densities of EGFP-Mlph-ΔRBD WT, E449A, and E452A were 1, 1.29, and 1.25, respectively. Correction of the transfection efficiencies of those three constructs (**A**; *Figure 6—figure supplement 1*) gave rise to the relative protein expression levels of each transfected cells as 1, 1.05, and 0.97 for EGFP-Mlph-ΔRBD WT, E449A, and E452A, respectively.

The online version of this article includes the following source data and figure supplement(s) for figure 6:

**Source data 1.** Original and statistical data for *Figure 6B*.

**Source data 2.** Original and uncropped blots for *Figure 6C*.

**Figure supplement 1.** Typical images of melanocytes expressing EGFP-Mlph-ΔRBD, its mutants, or EGFP.

Deletion of exon-F almost eliminated the co-localization of Myo5a-MTD and Mlph (*Figure 8B*), consistent with its essential role for the binding of Myo5a with Mlph. Deletion of exon-G substantially decreased, but did not eliminate, the co-localization of Myo5a-MTD and Mlph (*Figure 8C*), indicating that exon-G plays an auxiliary role for binding of Myo5a with Mlph.

## Discussion

It is well-established that Myo5a associates with melanosome via two independent interactions with Mlph, i.e., the interaction between exon-F of Myo5a and Mlph-EFBD and the interaction between the GTD and Mlph-GTBM. In the current study, we identified the third interaction between Myo5a and Mlph, i.e., the interaction between the exon-G of Myo5a and the Mlph-ABD. We demonstrated that, similar to the exon-F/EFBD interaction, the exon-G/ABD interaction is also required for the strong binding of Myo5a and Mlph.

Among the three interactions between Myo5a and Mlph, the GTD/GTBM interaction majorly regulates the motor function of Myo5a. At relative high ionic strength but not at physiological ionic strength, Mlph-GTBM binds to the GTD, inducing Myo5a to form the extended conformation and activating Myo5a motor activity (*Li et al., 2005*; *Yao et al., 2015*). At physiological ionic strength, the GTBM-mediated activation of Myo5a depends on the interaction between the GTD and RilpL2, which is regulated by Rab36 (*Cao et al., 2019*). Those findings are consistent with the three-dimensional structure of Myo5a-GTD and the folded structure of full-length Myo5a (*Niu et al., 2022*; *Pylypenko et al., 2013*; *Wei et al., 2013*), in both of which the GTBM-binding sites are buried at the GTD-GTD interface. Upon binding to the RH1 domain of RilpL2, the GTD exposes the GTBM-binding site (*Wei et al., 2013*). We therefore proposed that the binding of Rab36/RilpL2 to the GTD exposes the GTBM-binding site, thus facilitating GTBM to bind to the GTD and to activate the motor function of Myo5a (*Cao et al., 2019*).

The GTD/GTBM interaction is also required for the stable association between Myo5a and Mlph. The Myo5a tail construct containing all three Mlph-binding sites (the exon-F, the exon-G, and the GTD) produced dominant negative effect in melanocytes, and deleting the GTD abolished the dominant negative effect (*Figures 7 and 8*, and reference *Wu et al., 2002b*).

The exon-F/EFBD and the exon-G/ABD interactions play a major role in the binding of Myo5a with Mlph. Although exon-F and exon-G are adjacent to each other, the exon-F/EFBD interaction and the exon-G/ABD interaction do not interfere with each other. Rather, those two interactions act synergically. We observed that Myo5a-MTD, containing both exon-F and exon-G but lacking the GTD, did not generate a dilute-like phenotype in melanocytes, but was co-localized well with Mlph and partially with melanosomes. Deletion of either exon-F or exon-G substantially weakened the interaction between Myo5a-MTD and Mlph and decreased the colocalization of Myo5a-MTD with Mlph, indicating the both exon-F and exon-G are required for the colocalization.

Our observation that Myo5a-MTD, containing both exon-F and exon-G, was co-localized partially with melanosomes seems to contradict the early work from Hammer laboratory, which showed that MC STK (a Myo5a fragment contains both exon-F and exon-G) did not exhibit any tendency to co-localize with melanosomes (*Wu et al., 2002b*). However, careful examining their image (*Figure 5*, in reference

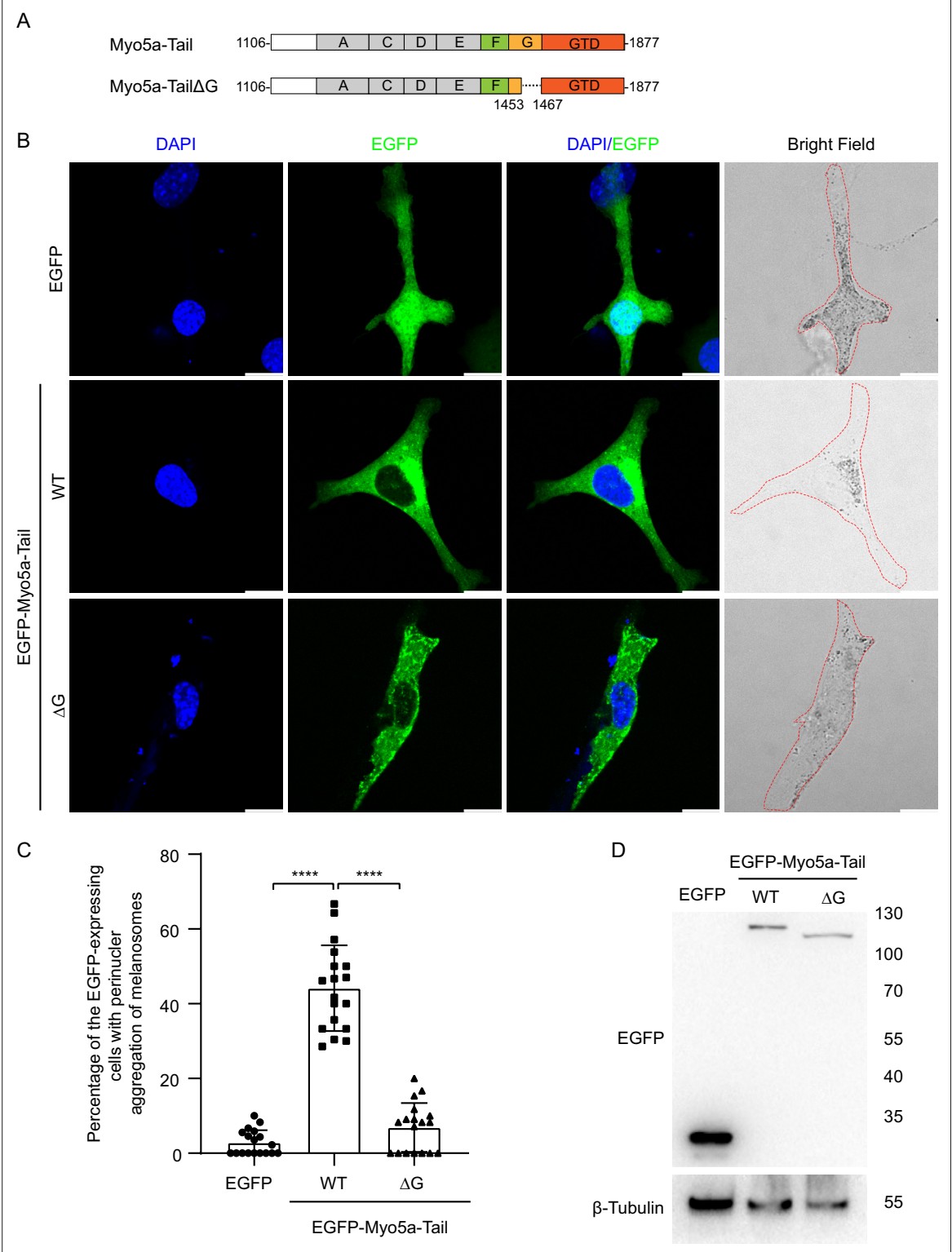

**Figure 7.** Exon-G region of Myo5a is essential for the perinuclear distribution of melanosomes induced by Myo5a-Tail overexpression. (**A**) Diagram of Myo5a-Tail constructs used for transfecting melan-a melanocytes. (**B**) Melan-a melanocytes were transfected to express EGFP-Myo5a-Tail and the mutant. The distribution of melanosomes in the transfected melanocytes was imaged and the number of melanocytes with perinuclear distribution of melanosomes was counted. Typical images of melanocytes expressing EGFP-Myo5a-Tail, its mutants, or EGFP. Cells are outlined with a dashed line. Scale bars = 10 μm. Zooms are x 3. (**C**) The percentages of melanocytes exhibiting perinuclear melanosome aggregation. Data are the mean ± SD of three independent experiments with one-way ANOVA with post-hoc Bonferroni test. ****$p<0.0001$. (**D**) Western blots of whole cell extracts prepared

*Figure 7 continued on next page*

*Figure 7 continued*

from melan-a melanocytes transfected with each of the three Myo5a constructs EGFP fusion protein constructs probed with an antibody against EGFP. Each point represents the percentages of perinuclear distribution of EGFP fusion proteins expressing cells in each fields. Western blot of the lysates of melan-a transfected with EGFP-Myo5a-Tail or EGFP-Myo5a-TailΔG. The expressed EGFP fusion proteins were probed with the antibody against EGFP. The relative band densities of EGFP-Myo5a-Tail and EGFP-Myo5a-TailΔG were 1 and 0.9, respectively. Correction of the transfection efficiencies of those three constructs (B; *Figure 7—figure supplement 1*) gave rise to the relative protein expression levels of each transfected cells as 1, 0.93 for EGFP-Myo5a-Tail or EGFP-Myo5a-TailΔG, respectively.

The online version of this article includes the following source data and figure supplement(s) for figure 7:

**Source data 1.** Original and statistical data for *Figure 7C*.

**Source data 2.** Original and uncropped blots for *Figure 7D*.

**Figure supplement 1.** Typical images of melanocytes expressing EGFP-Myo5a-Tail, its mutants, or EGFP.

Cells are outlined with a red dashed line. Scale bars = 10 μm. Zooms are x 0.75.

---

*Wu et al., 2002b*) reveals a partial colocalization of MC STK with melanosomes, particularly at the perinuclear region. It is likely that some melanosome-bound Mlph molecules are unoccupied, some interact with Myo5a via the exon-F/EFBD and the exon-G/ABD interactions but not the GTD/GTBM interaction, and some interact with Myo5a via three interactions. In the former two cases, Myo5a-MTD would be able to bind to the melanosome via Mlph. We therefore conclude that presence of both exon-F and exon-G is sufficient for Myo5a to associate with Mlph, but insufficient for substituting the melanosome-bound Myo5a molecules which bind to Mlph via three distinct interactions.

As discussed above, the GTBM-binding site is buried at the GTD-GTD interface in the folded state of Myo5a, thus is inaccessible for Mlph, and the opening of GTBM-site depends on the interaction with Rab36/RilpL2. On the other hand, the exon-F and the exon-G are likely exposed. We therefore propose that the folded Myo5a first attaches to Mlph via the exon-F/EFBD and exon-G/ABD interactions, then interacts with Rab36/RilpL2 to open the GTBM-site, and finally interacts with the GTBM, forming the extended conformation and being activated.

One interesting finding of this study is that Mlph-ABD was able to separately interact with the exon-G of Myo5a or actin filament, but unable to interact with both of them simultaneously. In other words, Mlph-ABD binds to either exon-G or actin filament. This is consistent with the geometry of Myo5a, in which exon-G is located far from the motor domain, which interacts with actin filament to produce motility. Given the small size of Mlph-ABD, it is unlikely that Mlph-ABD is able to bridge the exon-G at one end of Myo5a and actin filament which associates with the motor domain located at the other end of Myo5a. An unsolved issue is how Mlph-ABD's interactions with exon-G and actin filament are regulated. In vitro studies suggest that the interaction between Mlph-ABD and actin filament might be regulated by phosphorylation (*Oberhofer et al., 2017*).

Based on our current finding and previous studies on the interaction between Myo5a and its associated proteins, we propose following model for the Mlph-mediated Myo5a transportation of melanosome (*Figure 9*). At stage 1, Mlph associates with melanosome via its interaction with Rab27a, which directly binds to the membrane of melanosome; the unattached Myo5a is in a folded conformation, in which the GTD binds to and inhibits the motor domain. At stage 2, Mlph interacts with the folded Myo5a via the interactions of EFBD/exon-F and ABD/exon-G; the attached Myo5a is still in folded conformation, because the GTBM-binding surface in the GTD is buried at the GTD-GTD interface. At stage 3, the buried GTBM-binding surface between the GTD-GTD interface is exposed and thus facilitate the binding of GTBM, causing the dissociation of the GTD from the motor domain and inducing the extended conformation of Myo5a (This step is probably regulated by the binding of Rab36/RilpL2 to the GTD). At stage 4, Mlph-ABD dissociates from exon-G and then binds to actin filament, thus enhancing the processive movement of Myo5a (This step might be regulated by the phosphorylation of Mlph-ABD). The interaction between Mlph-ABD and actin filament is regulated by phosphorylation (*Oberhofer et al., 2017*).

Melanosomes in melanocytes are matured through a serial of well-defined morphological steps, from melanin-lacking premelanosomes to blackened melanosomes with melanin fully loaded (*Raposo and Marks, 2007*). During this maturation process, melanosomes undergo microtubule and actin-based transport towards the cell periphery, mediated by different Rab proteins and their effectors (*Fukuda, 2021*). For example, Rab27a mainly associates with intermediate and mature melanosomes

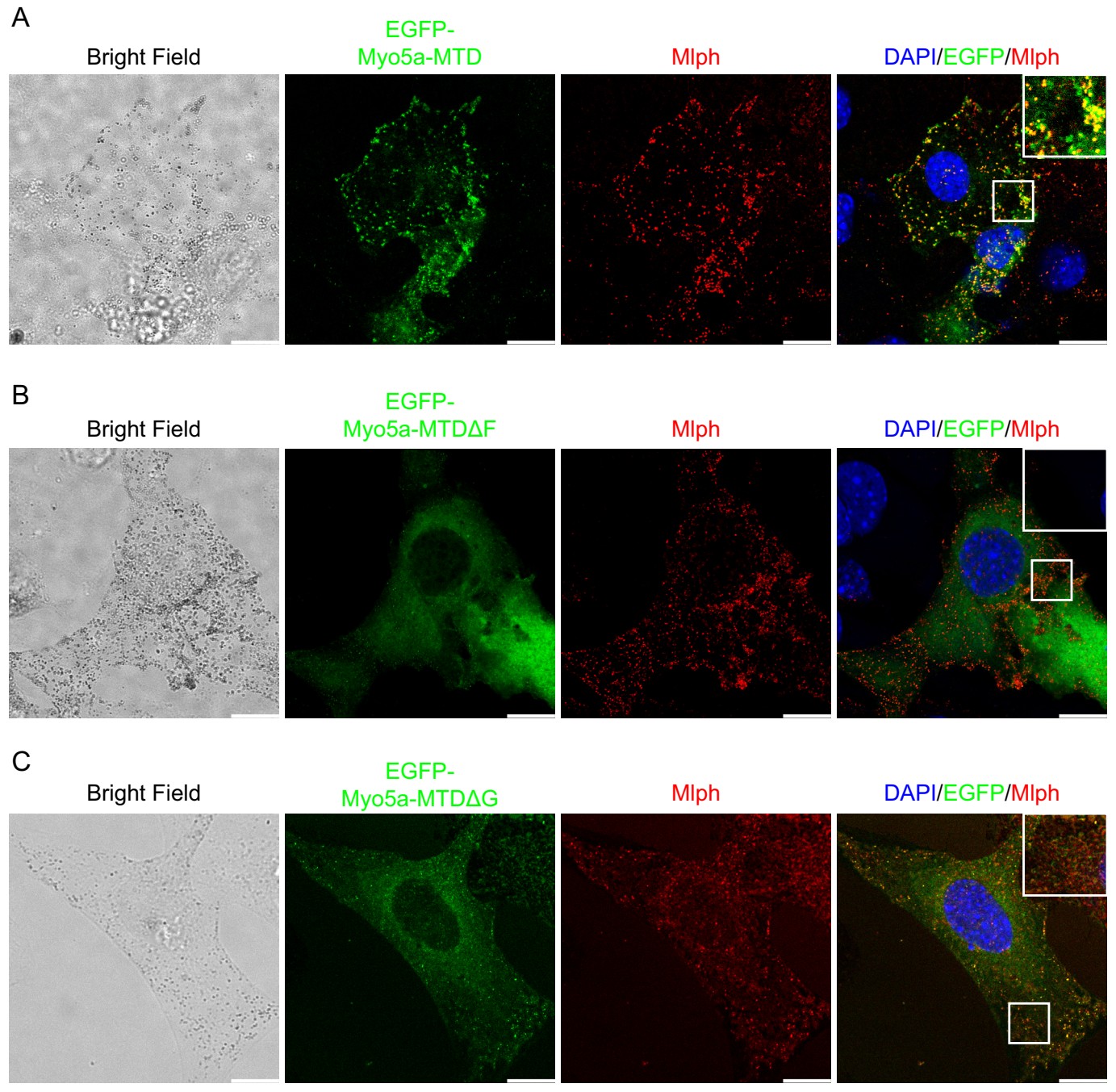

**Figure 8.** Effects of deletion of exon-F or exon-G on the localization of Myo5a-MTD in melan-a cells. Melan-a melanocytes were transfected to express EGFP-Myo5a-MTD (**A**), EGFP-Myo5a-MTDΔF (**B**), or EGFP-Myo5a-MTDΔG (**C**) and stained for endogenous Mlph. Insets represent higher magnification photomicrographs of a cell within the region outlined by frames. Scale bars = 10 µm.

(*Jordens et al., 2006*). The melanosome-bound Rab27a in GTP-bound state binds to Mlph, which in turn recruits Myo5a (*Wu et al., 2002a*). We propose that the recruitment of Myo5a by Mlph consists of multiple stages. We expect that Myo5a-MTD, containing both exon-F and exon-G, is able to bind to Mlph at stage 1 and 2, but not at stage 3 and 4, and that Myo5a-Tail, containing all three Mlph-binding sites, is able to bind to Mlph at all four stages.

In conclusion, we demonstrate a direct interaction between the exon-G of Myo5a and the ABD of Mlph, which is essential for the tight binding of Myo5a to Mlph both in vitro and in vivo. We expect that the melanosomal recruitment and activation of Myo5a are a highly coordinated process mediated

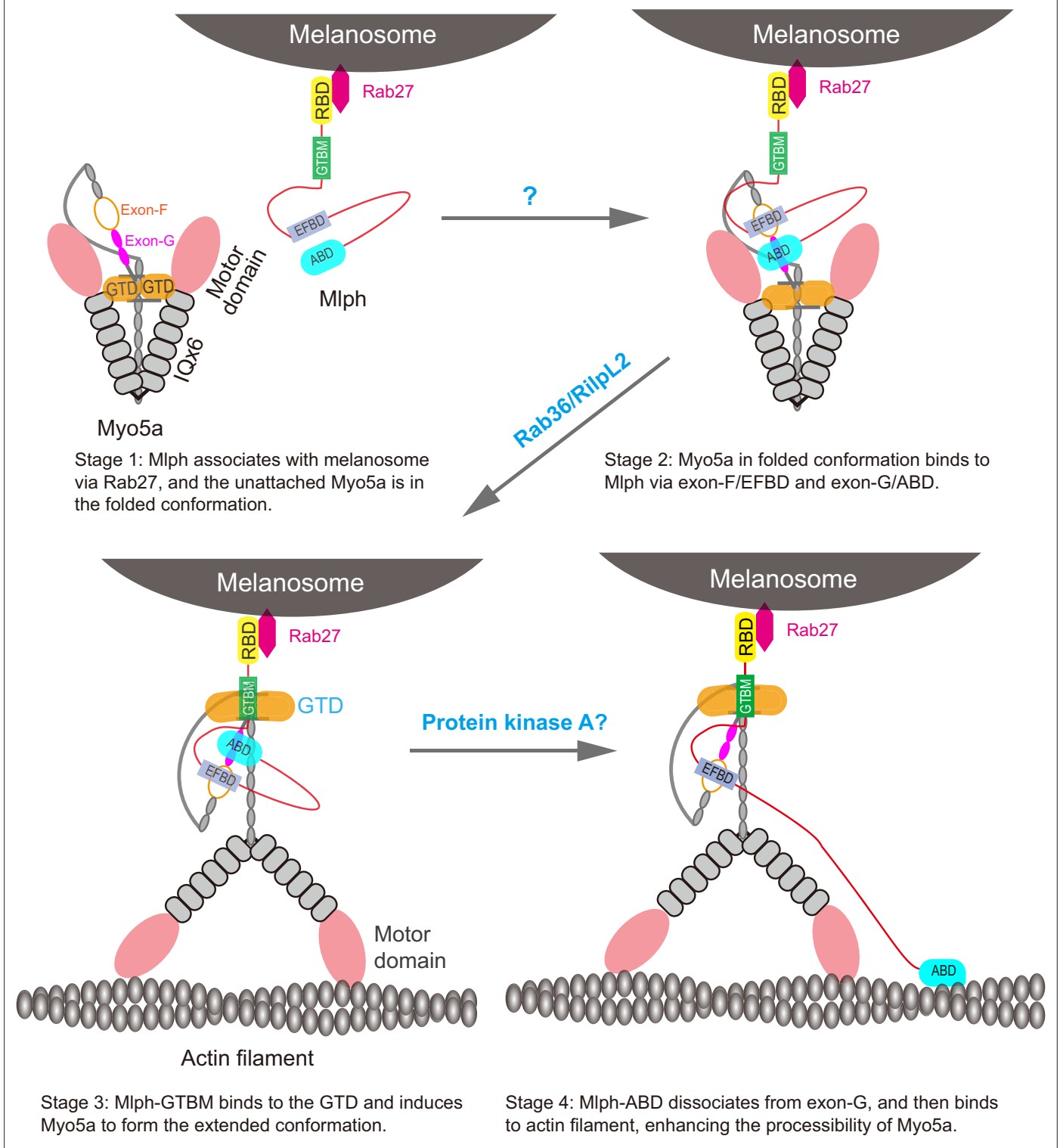

**Figure 9.** A model for the Mlph-mediated Myo5a transportation of melanosome. The Mlph-mediated Myo5a transportation of melanosome is comprised of four stages. At stage 1, Mlph associates with melanosome via its interaction with Rab27a, which directly binds to the membrane of melanosome; the unattached Myo5a is in a folded conformation, in which the GTD binds to and inhibits the motor domain. At stage 2, Mlph interacts with the folded Myo5a via the interactions of EFBD/exon-F and ABD/exon-G; the attached Myo5a is still in folded conformation, because the GTBM-binding surface in the GTD is buried at the GTD-GTD interface. At stage 3, the buried GTBM-binding surface between the GTD-GTD interface is exposed and thus facilitate the binding of GTBM, causing the dissociation of the GTD from the motor domain and inducing the extended conformation of Myo5a (This step is probably regulated by the binding of Rab36/RilpL2 to the GTD). At stage 4, Mlph-ABD dissociates from exon-G and then binds to actin filament, thus enhancing the processive movement of Myo5a (This step might be regulated by the phosphorylation of Mlph-ABD).

by three interactions between Myo5a and Mlph, that is the exon-F/EFBD interaction, the exon-G/ABD interaction, and the GTD/GTBM interactions.

## Materials and methods

### Materials

Restriction enzymes and modifying enzymes were purchased from New England Biolabs (Beverly, MA), unless indicated otherwise. Actin was prepared from rabbit skeletal muscle acetone powder according to *Spudich and Watt, 1971*. Ni-NTA agarose was purchased from Qiagen (Cat# 30250, Hilden, Germany). Anti-Flag M2 affinity agarose and HPR conjugated anti-Flag M2 antibody were from Sigma Co. (Cat# A8592, St. Louis, MO). EGFP (Cat# PTM-5180) and β-Tubulin (Cat# PTM-6414) antibody was purchased from PTMBio (Hangzhou, China). Glutathione(GSH)-Sepharose 4 Fast Flow (Cat# 17513201) was from GE Healthcare. FLAG peptide (DYKDDDDK) was synthesized by Augct Co. (Beijing, China). Oligonucleotides were synthesized by Invitrogen Co. (Beijing, China).

### Proteins

All Myo5a constructs in this study were created from a melanocyte type Myo5a (*Li et al., 2005*). The recombinant proteins of truncated Myo5a containing an N-terminal His-tag and Flag-tag were expressed in BL21(DE3) *E. coli*. The cDNA of truncated Myo5a were amplified by using high-fidelity FastPfu DNA polymerase (TransGen, Cat# AS211) with Myo5a cDNA as template (*Li et al., 2005*), and subcloned into pET30HFa (a modified pET30a vector encoding His-tag and Flag-tag). To improve the expression of short peptides of truncated Myo5a, including Myo5a(1411–1467) and Myo5a(1436–1467), the cDNA of Trx-1 (Genbank ID: P0AA25) was inserted between the Flag-tag sequence and the cDNA of truncated Myo5a in pET30HFa. Myo5a-MTDΔF were created by using overlap extension PCR with Myo5a-MTD/pFastFTb as template and inserted into pET30HFa. The recombinant proteins were expressed in BL21(DE3) *E. coli* as His-Flag tagged proteins were purified by Ni-agarose affinity chromatography using standard procedures. We also produced an N-terminal His-tagged and Flag-tagged Myo5a-MTD using baculovirus/Sf9 system. The cDNA of Myo5a-MTD was subcloned into the baculovirus transfer vector pFastHFTb (a modified pFastHTb vector containing the His-tag and Flag-tag sequence). Recombinant baculovirus was generated using Bac-to-Bac system. The Myo5a-MTD expressed in Sf9 insect cells was purified by Anti-FLAG M2 affinity chromatography (*Li et al., 2008*). The primers for creating Myo5a expression constructs are summarized in *Supplementary file 1*.

Three truncated Mlph constructs containing an N-terminal GST-tag, including Mlph-ABD (residues 401–590), Mlph-ΔRBD (residues 148–590), Mlph-ΔRBDΔABD (residues 148–400), were created by subcloning the cDNAs of truncated Mlph into pGEX4T1 vector using BamHI and XhoI sites. Mlph-EFBD (residues 241–400) having an N-terminal His-tag and Flag-tag was created by the cDNA of Mlph-EFBD subcloning into pET30HFa vector using EcoRI and HindIII sites. Point mutations were created using Fast Mutagenesis System (TransGen, Cat# AS231) according to the manufacturer's instructions. The recombinant proteins were expressed in BL21(DE3) *E. coli* as His-Flag tagged proteins (in pET30HFa vector) or GST tagged proteins (in pGEX4T1 vector), and purified by Ni-agarose affinity chromatography or GSH-Sepharose affinity chromatography using standard procedures. The primers for creating Mlph expression constructs are summarized in *Supplementary file 2*.

### Protein pulldown assay

GST pulldown assays were performed as described previously (*Zhang et al., 2016*). For GST pulldown of Myo5a-MTD with GST-Mlph-ABD, GSH-Sepharose beads (10 µl) were mixed with 95 µl of 2 µM GST-Mlph-ABD, 4 µM Myo5a-MTD truncations in Pulldown Buffer-I (5 mM Tris-HCl (pH 7.5), 100 mM NaCl, 1 mM DTT, and 1 mM EGTA, 0.1% NP-40) with rotation at 4 °C for 2 hr. The GSH-Sepharose beads were then washed three times with 200 µl of Wash Buffer-I (10 mM Tris-HCl (pH 7.5), 100 mM NaCl, 1 mM DTT, and 1 mM EGTA, 0.1% NP-40), before eluted by Elution Buffer (10 mM GSH, 50 mM Tris-HCl (pH 8.0), 1 mM DTT, and 200 mM NaCl). For GST pulldown of Myo5a-MTD with GST-Mlph-ΔRBD, GSH-Sepharose beads (10 µl) were mixed with 95 µl of 2 µM GST-Mlph-ΔRBD, 4 µM Myo5a-MTD in Pulldown Buffer-II (5 mM Tris-HCl (pH 7.5), 100 mM NaCl, 1 mM DTT, and 1 mM EGTA) with rotation at 4 °C for 2 hr. The GSH-Sepharose beads were washed three times with 200 µl of Wash

Buffer-II (10 mM Tris-HCl (pH 7.5), 100 mM NaCl, 1 mM DTT, and 1 mM EGTA) and then eluted by Elution Buffer.

For Flag pulldown assay, Flag-Myo5a-MTD (0.5 µM) was incubated with 1 µM GST-tagged Mlph truncations in Pulldown Buffer I and rotated at 4 °C for 2 hr, mixed with 10 µl of anti-FLAG M2 agarose. The anti-FLAG M2 agarose were washed three times with 200 µl of Wash Buffer I. The bound proteins were eluted twice with 20 µl of 0.1 mg/ml FLAG peptide in Wash Buffer I.

The inputs and the eluted proteins were analyzed by SDS-PAGE and visualized by Coomassie Brilliant Blue (CBB) staining or western blotting using the indicated antibody. The amounts of pulldowned proteins were quantified using ImageJ (version 1.42Q), and their molar ratios were calculated on the basis of their molecular masses.

## Microscale thermophoresis

The microscale thermophoresis (MST) assay of the interaction between Mlph-ABD with Myo5a-MTD or Myo5a-MTDΔG were carried out in a Monolith NT.115 instrument (NanoTemper Technologies, Germany) at 25 °C. Mlph-ABD was labelled with ATTO 488 NHS Ester dye (ATTO-TEC) according to the manufacturer's instructions. The Mlph-ABD was kept at a constant concentration of 20 nM. Two-fold dilution series (16 in total) of the Myo5a-MTD (50 µM) or Myo5a-MTDΔG (50 µM) were performed in the MST buffer (50 mM Tris-HCl (pH 7.8), 100 mM NaCl, 10 mM $MgCl_2$, 0.05% Tween 20). The dissociation constant $K_d$ was obtained using Monolith Affinity Analysis v2.2.4 software.

## Actin cosedimentation assay

Rabbit skeletal actin (2 µM) and Mlph-ABD (4 µM) was incubated in a 60 µL solution of 20 mM Tris-HCl (pH 7.5), 100 mM NaCl, 1 mM EGTA, and 1 mM DTT at 4 °C for 20 min with or without Myo5a-MTD (2 µM). In addition, Myo5a-MTD (0–8 µM) was incubated with rabbit skeletal actin (2 µM) and Mlph-ABD (1 µM) in a 60 µL solution of 20 mM Tris-HCl (pH 7.5), 100 mM NaCl, 1 mM EGTA, and 1 mM DTT at 4 °C for 20 min. The mixtures were centrifuged at 85,000 rpm (Beckman Optima MAX-XP, TLA-120.1 rotor) for 15 min at 4 °C. Equal portions of the pellet and the supernatant were subjected to SDS−PAGE and Coomassie brilliant blue staining.

## Melanophilin antibody

To generate a polyclonal antibody against melanophilin, the sequence encoding residues 147–428 of melanophilin (Mlph-147–428) was amplified by PCR using a full-length mouse melanophilin cDNA as template. The PCR product was cloned into pET30a vector and pGEX4T2 vector and expressed in *E. coli* as His-tagged protein (His-Mlph-147–428) and GST-tagged protein (GST-Mlph-147–428), respectively. His-Mlph-147–428 was purified using Ni-NTA agrose chromatography and used for immunizing rabbit. GST-Mlph-147–428 was purified using GSH-Sepharose chromatography. Mlph antibody was affinity-purified from immune sera over a column of GST-Mlph-147–428 coupled to cyanogen bromide-activated Sepharose 4B (GE Healthcare) by standard procedure.

## Plasmid constructions for melanocyte transfection

Myo5a-Tail/pEGFP-C1 was produced as described previously (**Wu et al., 2002b**). Myo5a-TailΔG/pEGFP-C1 was created by overlapping PCR using Myo5a-Tail/pEGFP-C1 as template. Myo5a-MTD/pEGFP-C1 and Myo5a-MTDΔG/pEGFP-C1 were produced by PCR amplification using Myo5a-Tail/pEGFP-C1 as template. Myo5a-MTDΔF/pEGFP-C1 was created by subcloning from Myo5a-MTDΔF/pET30a into pEGFP-C1. Mlph-ΔABD/pEGFP-C1 was created by subcloning Mlph-ΔABD cDNA from Mlph-ΔABD/pGEX4T1 into pEGFP-C1. Point mutations were introduced using Fast Mutagenesis System (TransGen, Cat# AS231) according to the manufacturer's instructions. All primes for creating transfection plasmids are summarized in **Supplementary file 3**.

## Melanocyte culture and transfection

The immortal mouse melanocyte cell line melan-a, kindly provided by Dr. Dorothy Bennett, was cultured as described (**Bennett et al., 1987**). Mycoplasma test is negative. Briefly, melan-a cells were cultured in RPMI 1640 medium (HyClone) supplemented with 10% fetal bovine serum, 200 nM phorbol 12-myristate 13-acetate and penicillin/streptomycin (1% v/v) in humidified chamber (37 °C, 5% $CO_2$ incubator). Melan-a cells were passaged at 3–4 day intervals. Hieff Trans Universal Transfection

Reagent (Yeasen) was used for transfection of pEGFP-C1 plasmids into melan-a cells according to the manufacturer's instructions.

## Immunocytochemistry

For immunofluorescence staining, melan-a cells were plated on the coverslips. After 24 hr culturing, cells were transfected of pEGFP-C1 plasmids. Two days after transfection, the cells were fixed using 4% paraformaldehyde for 20 min, then permeabilized with 0.4% Triton-X in PBS for 15 min. Subsequently, the coverslips were blocked with 1% BSA for 1 hr. Melan-a cells were incubated with rabbit antibody against Mlph overnight. Coverslips were washed with PBS for three times and then incubated for 1 hr at room temperature with secondary antibodies (anti-rabbit IgG coupled with DyLight 549, Jackson). Nuclei were counter-stained with DAPI. Coverslips were fixed on glass slides with Fluoromount-GTM (Yeasen). The images were captured by Leica STELLARIS 5 fluorescence microscope.

## Statistical analyses

All results are the mean ± SD of at least three independent experiments. Data among more than two groups was assessed with the parametric one-way ANOVA with post-hoc Bonferroni test. $*p < 0.05$, $**p < 0.01$, $***p < 0.001$ and $****p < 0.0001$, in comparison to control.

## Acknowledgements

We thank Dr. Zheng Zhou (Institute of Biophysics, Chinese Academy of Sciences) for advice on myosin-5a/melanophilin interaction. This work was supported by the National Natural Science Foundation of China (31970657).

## Additional information

### Funding

| Funder | Grant reference number | Author |
| --- | --- | --- |
| National Natural Science Foundation of China | 31970657 | Xiang-dong Li |

The funders had no role in study design, data collection and interpretation, or the decision to submit the work for publication.

### Author contributions

Jiabin Pan, Data curation, Formal analysis, Investigation, Visualization, Writing – original draft; Rui Zhou, Lin-Lin Yao, Ning Zhang, Investigation; Jie Zhang, Conceptualization, Investigation; Qing-Juan Cao, Resources; Shaopeng Sun, Methodology; Xiang-dong Li, Conceptualization, Data curation, Formal analysis, Supervision, Funding acquisition, Visualization, Writing – original draft, Writing – review and editing

### Author ORCIDs

Jiabin Pan ⓘ http://orcid.org/0000-0001-5055-6566
Xiang-dong Li ⓘ https://orcid.org/0000-0001-8677-9833

Reviewer #1 (Public Review): https://doi.org/10.7554/eLife.93662.3.sa1
Reviewer #2 (Public Review): https://doi.org/10.7554/eLife.93662.3.sa2
Author response https://doi.org/10.7554/eLife.93662.3.sa3

## Additional files

### Supplementary files

- Supplementary file 1. The primer sequences used in PCR for Myo5a constructs in pET30HFa.
- Supplementary file 2. The primer sequences used in PCR for Mlph constructs in pGEX4T1.

• Supplementary file 3. The primer sequences used in PCR for Myo5a or Mlph constructs in pEGFP-C1.

• MDAR checklist

### Data availability

All data generated or analysed during this study are included in the manuscript and supporting files; source data files have been provided for Figures 1,2,3,4,5,6,7 and Figure 2-figure supplement.

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
