## [Editor Report · eLife assessment]

This study represents a **useful** description of a third interaction site between melanophilin and myosin-5a which has a role in regulating the distribution of pigment granules in melanocytes. While much of the data forms a **solid** case for this interaction, the inclusion of controls for the cellular studies and measurement of interaction affinities would have been helpful.

---

## [Referee Report · Reviewer #1 (Public Review)]

Interactions known to be important for melanosome transport include exon F and the globular tail domain (GTD) of MyoVa with Mlph. Motivated by a discrepancy between in vitro and cell culture results regarding necessary interactions for MyoVa to be recruited to the melanosome, the authors used a series of pull-down and pelleting assays experiments to identify an additional interaction that occurs between exon G of MyoVa and Mlph. This interaction is independent of and synergistic with the interaction of Mlph with exon F. However, the interaction of the actin-binding domain of Mlph can occur either with exon G or with the actin filament, but not both simultaneously. These data lead to a modified recruitment model where both exon F and exon G enhance binding of Mlph to auto-inhibited MyoVa, and then via an unidentified switch (PKA?) the actin-binding domain of Mlph dissociates from MyoVa and interacts with the actin filament to enhance MyoVa processivity.

The only weakness noted is that the authors could have had a more complete story if they pursued whether PKA phosphorylation/dephosphorylation of Mlph is indeed the switch for the actin-binding domain of Mlph to interact with exon G versus the actin filament.

---

## [Referee Report · Reviewer #2 (Public Review)]

The authors identify a third component in the interaction between myosin Va and melanophilin- an interaction between a 32-residue sequence encoded by exon-g in myosin Va and melanophilin's actin binding domain. This interaction has implications for how melanosome motility may be regulated.

The authors have now included some necessary controls that were requested. In terms of adding new information to increase the significance and impact of the paper, they added a single affinity measurement. Unfortunately, it did not involve Exon G specifically. Moreover, they did not add any new mechanistic or functional data to provide a more conceptual advance. For example, is the Exon G interaction regulated by phosphorylation? Is this what dictates the choice between Mlph's actin binding domain (ABD) binding to actin or to exon-G. How does local actin concentration influence this decision. What changes regarding melanosome dynamics in cells between these two alternatives? Do in vitro reconstitution assays show that binding to Exon-G instead of actin affects the processivity of a Rab27a/Myosin 5a/Mlph transport complex? Finally, while the authors make clear in the abstract and text that they are just identifying a third component that mediates the Melanophilin-dependent association of myosin-5a with melanosomes, the title gives the impression that they identified all three in this manuscript. I really think the title should be changed to something like Identification of a third component that mediates the Melanophilin-dependent association of myosin-5a with melanosomes, as this accurately reflects what is new in this work.

---

## [Author Response]

The following is the authors’ response to the original reviews.

We appreciate your comments and suggestions on our manuscript.

In particular, we have measured the affinity between the middle tail domain of myosin-5a (Myo5a-MTD) and the actin-binding domain of melanophilin (Mlph-ABD) using microscale thermophoresis, and obtained the Kd of ~0.56 uM, which is similar to the Kd of the globular tail domain of myosin-5a (Myo5a-GTD) to the GTD-binding motif of melanophilin (Mlph-GTBM). Moreover, we have performed Western blot of the lysate of transfected cells, showing that the proteins of the dominant negative construct and the negative control were expressed at similar lever without noticeable degradation.

We appreciate the editors’ and reviewers’ comment on how melanophilin might be regulated in binding to the exon-G of myosin-5 and to actin filaments. Phosphorylation of melanophilin by protein kinase A is one possible mechanism. We will investigate this issues in our future study.

We also took this opportunity to correct several minor errors in the manuscript. Textual alterations can be viewed in the “tracked change” version of the manuscript. Below is the comments from the editors and the two reviewers together with our point-by-point responses.

**eLife assessment**
This study represents a useful description of a third interaction site between melanophilin and myosin-5a which is important in regulating the distribution of pigment granules in melanocytes. While much of the data forms a solid case for this interaction, the inclusion of important controls for the cellular studies and measurement of interaction affinities would have been helpful.
**Public Reviews:**

**Reviewer #1 (Public Review):**
Interactions known to be important for melanosome transport include exon F and the globular tail domain (GTD) of MyoVa with Mlph. Motivated by a discrepancy between in vitro and cell culture results regarding necessary interactions for MyoVa to be recruited to the melanosome, the authors used a series of pull-down and pelleting assays experiments to identify an additional interaction that occurs between exon G of MyoVa and Mlph. This interaction is independent of and synergistic with the interaction of Mlph with exon F. However, the interaction of the actin-binding domain of Mlph can occur either with exon G or with the actin filament, but not both simultaneously. These data lead to a modified recruitment model where both exon F and exon G enhance the binding of Mlph to auto-inhibited MyoVa, and then via an unidentified switch (PKA?) the actin-binding domain of Mlph dissociates from MyoVa and interacts with the actin filament to enhance MyoVa processivity.The only weakness noted is that the authors could have had a more complete story if they pursued whether PKA phosphorylation/dephosphorylation of Mlph is indeed the switch for the actin-binding domain of Mlph to interact with exon G versus the actin filament.

We thank Reviewer #1 for careful reading of the manuscript and appreciation of the study. We agree with the Reviewer that it is important to understand how the actin-binding domain of Mlph switch its interaction with the exon-G of Myo5a and actin filament. We would like to pursue this direction in our future research.

**Reviewer #2 (Public Review):**
The authors identify a third component in the interaction between myosin Va and melanophilin- an interaction between a 32-residue sequence encoded by exon-g in myosin Va and melanophilin's actin-binding domain. This interaction has implications for how melanosome motility may be regulated.While this work is largely well done and certainly publishable following needed revisions (e.g. some affinity measurements, necessary controls for the dominant negative experiments), I believe that additional work would be required to make a more compelling case. First, the study provides just one more piece to a well-developed story (the role of exon-F and the GTD in myosin Va: melanophilin (Mlph) interaction), much of which was published 20 years ago by several labs. Second, the study does not demonstrate a physiological significance for their findings other than that exon-G plays an auxiliary role in the binding of myosin Va to Mlph. For example, what dictates the choice between Mlph's actin binding domain (ABD) binding to actin or to exon-G. Is it a PTM or local actin concentration? It is unlikely to be alternative splicing as exon-G is present in all spliced isoforms of myosin Va. And what changes re melanosome dynamics in cells between these two alternatives? Similarly, the paper does not provide any in vitro evidence that binding to exon-G instead of actin effects the processivity of a Rab27a/Myosin Va/Mlph transport complex. For example, if the ABD sticks to exon-G instead of actin, does that block Mlph's ability to promote processivity through its interaction with the actin filament during transport? In summary, given that the authors did not directly test their model either in vitro or in cells, I do not think this story represent a significant conceptual advance.

We thank Reviewer #2 for careful reading of the manuscript and the suggestions of improving the manuscript. As suggested by the reviewer, we have measured the affinity between the middle tail domain of Myo5a (Myo5a-MTD) and Mlph-ABD (Kd ~0.562 uM), which is similar to that between the globular tail domain of Myo5a (Myo5a-GTD) and the GTBM of Mlph. In addition, we have performed additional experiments showing the integrity and the expression level of the dominant negative constructs in the transfected cells.

We believe more extensive experiments are required to address other questions raised by the reviewer. For example, what dictates the choice between Mlph's actin binding domain (ABD) binding to actin or to exon-G is an open question. As we proposed, phosphorylation by protein kinase A is only one possible mechanism. We would like to pursue them in our future research.

**Recommendations for the authors:**
The reviewing editor feels strongly that addressing some of the points raised by the reviewers would make this a more compelling manuscript. In particular, a measurement of the affinity of the relevant fragments from melanophilin and myosin-5a would indicate that the interaction might be physiologically relevant. Concerning the dominant negative experiments, the lack of effect of an expressed fragment could be that the expressed fragments were simply degraded or expressed at too low of a level to be competing. The reviewer gives guidelines on how to address this. Reviewer #2 made a point that it would be compelling if the effect of phosphorylation as suggested in the model was tested, but we all agree that this could well be the subject of a later study. In addition, the authors make a very interesting proposal for how protein kinase A could be involved in this regulation as has been suggested previously. Perhaps the use of phosphomimetic mutations could give some insight into this. Such experiments, if consistent with the proposed model would certainly raise the impact of this study. Finally, a very clear periodicity in hydrophobic amino acids is apparent in the interacting sequences of both Myo5 (yrisLykrMidLmeqLekqdktVrkLkkqLkvFakkIgeLevgqmen) and Mlph (tdeeLseMedrVamtAseVqqAeseIsdIesrIaaLra). This is strongly suggesting a leucine-zipper-like coiled coil, rather than an interaction mediated solely by charge. Recent softwares (and easily accessible too) like AlphaFold multimer might yield important structural insight into the binding configuration and might help rationalize the effect of the mutations herein.

We thank the editors and the reviewers for their suggestions of improving the manuscript. We have performed the several essential experiments to address the concerns raised by the reviewers.

(1) Regarding the affinity of the relevant fragments from melanophilin and myosin-5a. We have measured the affinity between Mlph-ABD and Myo5a-MTD using MST (Kd ~562 nM) (see revised Figure 3A).

(2) Regarding the concerns on the dominant negative experiments. We have examined the molecular sizes and expression levels of Mlph or Myo5a constructs by Western blots. First, we show that all constructs have correct molecular size in transfected cells (see revised Figure 6C and 7D), indicating that the inability of Myo5a or Mlph truncations to generate dilute-like phenotypes was not due to the intracellular degradation of the EGFP fusion protein. Second, by correcting for the percentage of transfected cells, we show that the overall expression levels of the wild-type construct and the mutants are roughly equal. Third, we categorized the expression levels into high and low, and calculated percentage of the DN phenotype in high and low expression levels. The results are consistent with the percentage of DN phenotype in total EGFP fusion protein cells.

(3) Regarding the suggestion to investigate the effect of phosphorylation by protein kinase A on Mlph-ABD’s interaction with Myo5a and actin filament. We understand that it is important to elucidate the mechanism by which the actin-binding domain of Mlph switch its interaction with the exon-G of Myo5a and actin filament. However, as we proposed, phosphorylation by protein kinase A is one possible mechanism, and more extensive experiments are required to address this question. Therefore, we would like to pursue it in our future research.

(4) Regarding the suggestion to predict the interaction between the exon-G of myosin-5a and Mlph-ABD using AlphaFold. We have used AlphaFold multimer to predict the Myo5a-MTD/Mlph-ABD interaction. Remarkably, the AlphaFold predicted that the binding of Myo5a-MTD with Mlph-ABD is mediated by an antiparallel coiled-coil formed by Myo5a (1430-1467) and Mlph (450-481), just as predicted by the editors. This prediction is also consistent with our finding that the exon-G of Myo5a interacts with Mlph-ABD. However, the predicted model cannot explain our mutagenesis results. We will pursue this point in the future research. Nevertheless, we are grateful to the editors for bringing this idea to our attention, because it will help us to design experiments to investigate the nature of Myo5a-exon-G/Mlph-ABD interaction.

**Reviewer #1 (Recommendations For The Authors):**
Specific minor commentsQ1: In figs 6-7 an overlay between DAPI and EGFP would be helpful for the reader to see perinuclear distribution.

As suggested, we have added the merged images of DAPI and EGFP in the revised Figure 6 and 7.

Q2: The delta symbol in the pdf text was corrupted.

The corrupted delta symbol has been fixed in the revised manuscript.

**Reviewer #2 (Recommendations For The Authors):**
Q1: Please explain in detail early in the text what exon-G is - length, position in the tail, and evidence that it is a coiled coil (CC). Of note, is it only long enough for about 4 heptad repeats? Has it been shown biochemically to form a CC? Is the CC irreversible? What would be the consequence of removing the exon-G CC on the ability of surrounding regions to bind Mlph (exon-F and the GTD)?

We thank the reviewer for this suggestion. In the revision, we added a new paragraph (the first paragraph in the results section) and revised Figure 1A to introduce the middle tail domain and alternatively spliced exons of Myo5a.

Exon-G is 32 amino acids in length, located at the C-terminal region of the middle tail domain, immediately before the globular tail domain. Exon-G region was predicted to form a short coiled-coil by using on-line tools (such as paircoil), and this prediction has not been tested biochemically. Moreover, we do not know whether the exon-G coiled-coil is reversible or not.

We have not examined the effect of removing the whole exon-G on the interaction between the GTD and Mlph-GTBM. The exon-G (residues 1436-1467) and the GTD core (residues 1498-1877) are separated by a long loop of 31 residues. We therefore expect that the removing the exon-G will not affect the GTD/Mlph-GTBM interaction.

Physically, exon-F is immediately followed by exon-G, and those two regions might interfere with each other. In our preliminary study, we found that removing the whole exon-G abolished the interaction between exon-F and Mlph-EFBD. On the other hand, removing the C-terminal half (residues 1454-1467) of exon-G had little effect the interaction between exon-F and Mlph-EFBD (see Figure 2C). In this work, we intentionally selected the later construct for functional analysis of the exon-G/Mlph-ABD interaction, because removing the C-terminal half of exon-G abolishes the interaction with Mlph-ABD, but does not affect the exon-F/Mlph-EFBD interaction.

Q2: Figures 1-3. While the pulldown experiments demonstrating an interaction between Mlph-ABD residues 446-571 and Myo5a-MTD are a good start, one would like to see affinity measurements to gauge the likelihood that this interaction is physiologically relevant. The same goes for the pulldown experiments demonstrating an interaction between (i) the C-terminal half of exon-G (residues 1453-1467) and the Mlph-ABD, (ii) between residues 1411-1467 (a short peptide containing exon-F and exon-G) and the Mlph-ABD, and (iii) between residues 1436-1467 (a short peptide containing exon-G) and the Mlph-ABD. This would also apply to the pulldowns in 3C-3E where versions of the proteins with charge residue changes were tested.

We agree the reviewer’s opinion that determination of the affinities between Mlph-ABD and Myo5a-MTD and their variants will be helpful in understanding the physiological relevance of Exon-G/Mlph-ABD interaction. However, the extensive experiments suggested by the reviewer require many high quality, purified proteins, which are not trivial.

Nevertheless, we think it is important to know the affinity between Myo5a-MTD and Mlph-ABD (both wild-type), as this parameter can be used for the comparison of the three interactions between Myo5a and Mlph. Therefore, we have obtained the affinity between Myo5a-MTD and Mlph-ABD using microscale thermophoresis (MST). The dissociation constant (Kd) of Myo5a-MTD to Mlph-ABD is 0.562±0.169 uM, which is similar to that between Myo5a-GTD and Mlph-GTBM (~1 uM) (Geething & Spudich (2007) JBC 282:21518). Consistent with GST pulldown results, MST shows that deletion of C-terminal half of exon-G (1453-1467) greatly decreases the MST signals (see revised Figure 3A).

Q3: While the domain negative (DN) approach to testing functional significance is OK, rescuing dilute/myosin Va null melanocytes with full-length myosin Va containing the various deletions would have been more convincing. Also, the authors must show (i) that the DN constructs are the correct size in transfected cells (i.e. are not degraded), and (ii) that they are expressed at roughly equal levels (either by doing Westerns and correcting for the percent of transfected cells, or by measuring total cellular fluorescence in transfected cells). Without this information, it remains possible that constructs not exhibiting a DN effect are simply degraded or poorly expressed. This applies to all the DN data in Figures 6 and 7.

We agree with the reviewer that Myo5a null melanocytes is ideal for investigating exon G function. Unfortunately, we do not have Myo5a null melanocytes derived from dilute mice.

To confirm the integrity of the overexpressed proteins in the transfected cells, we performed Western blot of those proteins, including EGFP-Mlph-ΔRBD (wild-type and two mutants) and Myo5a-Tail (wild-type and ΔG mutant), in the lysate of the transfected cells. Western blots show that all those proteins have correct molecular masses, indicating no degradation of those overexpressed proteins (see revised Figure 6C and 7C). Moreover, by correcting for the percentage of transfected cells, we show that the overall expression levels in each transfected cell of the wild-type construct and the mutants are roughly equal. This information is included in the revised manuscript (Line 222-225; 237-241).

Q4: The authors scored the DN phenotype as yes/no but it mostly likely varies depending on the degree of over-expression. Showing that the degree of melanosome centralization scales with the degree of overexpression, and that the correlation between expression level and phenotype varies depending on the construct would strengthen the results.

We agree with the reviewer’s prediction that the degree of DN phenotype should depend on the of over-expression level. We analyzed the EGFP signals of transfected cells and found very few cells with medium expression level. Therefore, we simply categorized the expression levels into high and low, and calculated the DN phenotype in each categories as shown in the table below. These results are consistent with the expectation that the degree of DN phenotype depends on the over-expression level of the transfected constructs.

**Author response table 1. sa3table1:** Percentage of the EGFP-expressing cells with perinuclear aggregation of melanosomes.

Transfected construct	High expression	Low expression
EGFP-Mlph-ΔRBD WT	59.2%(n=94)	23.8%(n=74)
EGFP-Mlph-ΔRBD E449A	22.5%(n=103)	4.9%(n=71)
EGFP-Mlph-ΔRBD E452A	20.8%(n=120)	3.7%(n=77)
EGFP-Myo5a-Tail	69.3%(n=96)	11.9%(n=63)
EGFP-Myo5a-TailΔG	16.7%(n=110)	2.1%(n=72)

Q5: The conclusion from the data in Figure 8A- "the presence of both exon-F and exon-G is insufficient for binding to the Mlph occupied by Myo5a, but sufficient for binding to the unoccupied Mlph"- should be verified by also doing the experiment in myosin Va knockdown cells.

We agree. Unfortunately, our RNAi knockdown of Myo5a in melanocytes by RNAi is not ideal and we do not have Myo5a knockout melanocytes. We will pursue this point in the future.

Q6: Line 213 "three Mlph-binding regions, i.e., exon-F, exon-F, and GTD (Figure 7A)" has a typo.

This typo has been corrected.

Q7: The authors should provide high mag insets for the images in Figure 8.

As suggested, we have revised Figure 8 by including high mag insets for the images.